# Neural Bootstrapping Attention for Neural Processes

## Abstract

Neural Processes learn to fit a broad class of stochastic processes with neural networks. Modeling functional uncertainty is an important aspect of learning stochastic processes. Recently, Bootstrapping Neural Processes (B(A)NP) propose a bootstrap method to capture the functional uncertainty which can replace the latent variable in (Attentive) Neural Processes ((A)NP), thus overcoming the limitations of Gaussian assumption on the latent variable. However, B(A)NP conduct bootstrapping in a non-parallelizable and memory-inefficient way and fail to capture diverse patterns in the stochastic processes. Furthermore, we found that ANP and BANP both tend to overfit in some cases. To resolve these problems, we propose an efficient and easy-to-implement approach, Neural Bootstrapping Attentive Neural Processes (NeuBANP). NeuBANP learns to generate the bootstrap distribution of random functions by injecting multiple random weights into the encoder and the loss function. We evaluate our models in benchmark experiments including Bayesian optimization and contextual multi-armed bandit. NeuBANP achieves the best performance in the sequential decision-making tasks among NP methods, and this empirically shows that our method greatly improves the quality of functional uncertainty modeling.

## 1 Introduction

Neural Processes (NP) (Garnelo et al., 2018b) define distributions over functions given a set of observations, and are trained via a meta-learning framework so that it can adapt to new functions rapidly. NP learns to model a wide range of stochastic processes and can estimate the uncertainty over the predictions with less computational effort, compared to Gaussian Processes (GP) (Rasmussen, 2003). However, NP frequently suffers from a fundamental drawback of underfitting. As a remedy of the underfitting issue, Attentive Neural Processes (ANP) (Kim et al., 2018) applies the attention modules to the encoder network. Despite this modification, a single Gaussian latent variable of (A)NP has a limitation in inducing *functional uncertainty* (Louizos et al., 2019), a global uncertainty that decides the distribution over the space of trajectories or functions.

Appropriate modeling of functional uncertainty in stochastic processes improves the predictive performance and diversity in function realizations (Le et al., 2018), thus provides a principled way to guide agents to find optimal candidates in sequential decision-making problems. In these tasks, a model needs to approximate a function and estimate uncertainty correctly to optimize a black-box function whose analytic information is not given. Although GP is widely used for these tasks, these are the promising area for the application of NP because GP is computationally expensive, and it can be hard to choose an appropriate prior. Recently, Bootstrapping Neural Processes (B(A)NP) (Lee et al., 2020) modify (A)NP to induce more robust uncertainty estimation by employing the residual bootstrapping. However, B(A)NP underperforms in capturing a functional uncertainty because the residual bootstrapping works in a homoscedastic way, removing the connection between the feature and the label in its bootstrapped samples. The bootstrap strategy used in B(A)NP demands a higher computational burden compared to (A)NP since it requires multiple computations of the encoder network and additional heuristics, including the adaptation layer and the lower bound on the variance[1]. Furthermore, ANP and BANP tend to overfit for a simple regression task rather than learning the underlying heteroscedasticity. We observed that this problem is a by-product of the attention modules

---

[1]We colored the revised or added sentences in blue only for the rebuttal. This color will not appear in the final version of the manuscript.

used in both models (see Figure 1 and 5). This finding suggests that effective regularization of attention modules is required to prevent overfitting. Additionally, as explained in Section 4, ANP and BANP often tend to estimate homogeneous uncertainty regardless of whether the observation is given.

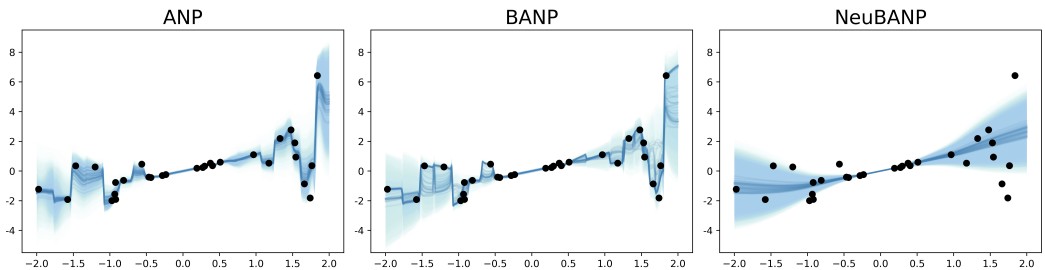

**Fig. 1.** Each plot shows predictions given by ANP, BANP, and the proposed method in a linear regression. The ground-truth function is a simple linear function with heterogeneous variance: $y = x + \beta\epsilon(x)$ where $\epsilon(x) \sim N(0, \sigma^2(x))$ and $\sigma(x) = \sqrt{x^2 + 10^{-5}}$. See Appendix B.1 for more details.

To resolve the problems of previous NPs, we introduce a novel bootstrapping method for neural processes, **Neu**ral **B**ootstrapping **A**ttentive **N**eural **P**rocesses (NeuBANP). Motivated from the recent work on efficient bootstrapping of the neural network, Neural Bootstrapper (NeuBoots) (Shin et al., 2021), we introduce bootstrapping of the attention in a computationally efficient way by simple modification of the input of attention modules and the loss function, instead of memory-inefficient resampling and contrived heuristics employed in BANP. The simplicity and computational efficiency of NeuBANP directly come from NeuBoots, but it does not guarantee the performance in modeling the functional uncertainty since NeuBANP operates on the meta-learning framework. Thus, we modify the method for training the model to learn the randomness present in the underlying function, allowing the model to estimate random functions generated by any stochastic process. This modification is simple but gives a strong consistency between the bootstrapped samples and the representations from the attention modules, unlike the residual bootstrapping used in BANP. Besides, our bootstrapping method, which implements the concatenation and multiplication of random bootstrap weights, operates as a regularizer on the attention networks, thus preventing overfitted predictions observed in previous attention-based NP methods. NeuBANP is trained to generate a valid predictive distribution by utilizing the uncertainty inherent in observations by bootstrapping instead of the uncertainty that depends on the prior assumption on the latent variable as in (A)NP. This leads to the success in capturing heteroscedasticity of the data and modeling functional uncertainty in stochastic processes. As a result, NeuBANP achieves the best performance in sequential decision-making problems such as Bayesian optimization and contextual multi-armed bandit. The experimental results demonstrate that our model provides promising capabilities as an efficient neural approximation of stochastic processes.

**Contributions**   We propose NeuBANP, an easy-to-implement and computationally efficient method for bootstrapping ANP. The proposed method has a novelty in learning a generator for bootstrapping stochastic processes under a meta-learning framework. NeuBANP resolves overfitting problem of attention modules and shows robust performance on heteroscedastic models without heuristics like the extra adaptation layer in BANP. NeuBANP estimates functional uncertainty better than BANP and achieves the best performance in stochastic optimization problems, including multi-dimensional Bayesian optimization and contextual multi-armed bandit, compared to previous NP methods.

## 2   PRELIMINARIES

### 2.1   META-LEARNING FRAMEWORK OF NEURAL PROCESSES

Consider data $\mathcal{D} = (X, Y) = \{(x_i, y_i)\}_{i=1}^n \subset \mathcal{X} \times \mathcal{Y}$, the pairs of inputs $x_i \in \mathcal{X}$ and outputs $y_i \in \mathcal{Y}$. Let $\mathcal{P}$ be a probability distribution over functions $f \in \mathcal{F}$; $y_i = f(x_i) + \epsilon_i$ where $\epsilon_i \sim \mathcal{N}(0, \sigma_i^2)$, hence $\mathcal{P}$ determines the distribution of $\mathcal{D}$. For disjoint index sets $\mathcal{C}$ and $\mathcal{T}$ satisfying $\mathcal{C} \cup \mathcal{T} = [n]$, define context $\mathcal{D}_\mathcal{C} = (X_\mathcal{C}, Y_\mathcal{C}) = \{(x_c, y_c)\}_{c \in \mathcal{C}}$ and target $\mathcal{D}_\mathcal{T} = (X_\mathcal{T}, Y_\mathcal{T}) = \{(x_t, y_t)\}_{t \in \mathcal{T}}$, so that $\mathcal{D} = \mathcal{D}_\mathcal{C} \cup \mathcal{D}_\mathcal{T}$. The task is to learn the neural processes $p_\theta$ that fits the stochastic processes

$f \sim \mathcal{P}$ given $\mathcal{D}_\mathcal{C}$ when $\mathcal{C} \sim \mathcal{P}_n$ is a randomly chosen subset of $[n]$, as follows:

$$\theta_\star = \underset{\theta}{\operatorname{argmin}} \, \mathbb{E}_{f \sim \mathcal{P}} \Big[ \mathbb{E}_{\mathcal{C} \sim \mathcal{P}_n} \big[ -\log p_\theta(Y|X, \mathcal{D}_\mathcal{C}) \big] \Big]. \tag{1}$$

This meta-learning framework allows $p_\theta$ to learn diverse patterns in $\mathcal{F}$ via a prior distribution $\mathcal{P}$; hence NP can predict target points conditioned on contexts adaptively in the inference phase.

## 2.2 (BOOTSTRAPPING) ATTENTIVE NEURAL PROCESSES

**ANP** ANP is a variant of NP equipped with attention modules in the encoder part. See Appendix A for the detailed definition of attention operations. Let the context $\mathcal{D}_\mathcal{C}$ is given, and the model aims to infer $y$ for a given feature $x \in \mathcal{X}$. The encoder network of ANP maps $(x, \mathcal{D}_\mathcal{C})$ into a pair of representation vectors $\mathbf{r} = (\mathbf{z}, \mathbf{h})$ as follows:

$$\{s_c\}_{c \in \mathcal{C}} = \text{SelfAttn}(\mathcal{D}_\mathcal{C}), \quad \mathbf{s}_\mathcal{C} = \text{mean}(\{s_c\}_{c \in \mathcal{C}}), \quad \mathbf{z} \sim \mathcal{N}(z|\mu_z(\mathbf{s}_\mathcal{C}), \sigma_z^2(\mathbf{s}_\mathcal{C})) \tag{2}$$

$$\{h_c\}_{c \in \mathcal{C}} = \text{SelfAttn}(\mathcal{D}_\mathcal{C}), \quad \mathbf{h} = \text{CrossAttn}(x, X_\mathcal{C}, \{h_c\}_{c \in \mathcal{C}}) \tag{3}$$

Here, $\mu_z$ and $\sigma_z$ are single linear layers that map $\mathbf{s}_\mathcal{C}$ to mean and standard deviation of the latent variable $\mathbf{z}$, respectively. In ANP, $\mathbf{z}$ and $\mathbf{h}$ refer to latent path and deterministic path, respectively. The deterministic path models the overall skeleton of the encoder network, while the latent path models the functional uncertainty using a stochastic global latent variable (Garnelo et al., 2018b; Kim et al., 2018). Then the decoder network takes the representations $\mathbf{r}$ and target data $x$ as inputs to predict $\mu(x, \mathbf{r})$ and $\sigma(x, \mathbf{r})$, the parameters of the conditional predictive distribution $p(y|x, \mathcal{D}_\mathcal{C}) = \mathcal{N}(y|\mu, \sigma^2)$.

**BANP** The global latent variable in ANP potentially limits the flexibility in expressing functional uncertainty. BANP proposes a method that utilizes paired bootstrapping and residual bootstrapping to model stochasticity in a data-driven way. First, they resample pairs of $(x_c, y_c)$ with replacement to construct $\hat{\mathcal{D}}_\mathcal{C}^{(b)} = (\hat{X}_\mathcal{C}^{(b)}, \hat{Y}_\mathcal{C}^{(b)})$. Here $b = 1, \ldots, B$ implies the number of bootstrap samples. The resampled context is encoded into the representation vector $\hat{\mathbf{r}}^{(b)} = (\hat{\mathbf{z}}^{(b)}, \hat{\mathbf{h}}^{(b)})$ by replacing $\mathcal{D}_\mathcal{C}$ in (2) and (3) by $\hat{\mathcal{D}}_\mathcal{C}^{(b)}$. Then BANP conducts the residual bootstrapping to construct bootstrapped contexts again $\tilde{\mathcal{D}}_\mathcal{C}^{(b)} = (\tilde{X}_\mathcal{C}^{(b)}, \tilde{Y}_\mathcal{C}^{(b)})$. By using $\mathcal{D}_\mathcal{C}$ and $\tilde{\mathcal{D}}_\mathcal{C}^{(b)}$ in (2) and (3) separately, BANP gets the representations of contexts $(\mathbf{r}, \tilde{\mathbf{r}}^{(b)})$. Finally, BANP uses an adaptation layer to merge $(\mathbf{r}, \tilde{\mathbf{r}}^{(b)})$ and obtain $\mu^{(b)}, \sigma^{(b)}$ through the decoder network (see Figure 2).

$$p(y|x, \mathbf{r}, \tilde{\mathbf{r}}^{(b)}) = \mathcal{N}(y|\mu^{(b)}, (\sigma^{(b)})^2), \quad p(y|x, \mathcal{D}_\mathcal{C}) \approx \frac{1}{B} \sum_{b=1}^{B} p(y|x, \mathbf{r}, \tilde{\mathbf{r}}^{(b)}). \tag{4}$$

Due to the residual bootstrapping, BANP conducts the encoder computation three times and the decoder computation twice for a single forward propagation. These additional calculations cause the computational bottleneck in the training and inference (see Appendix B.6).

## 2.3 NEURAL BOOTSTRAPPER

Repetitions of training restrain the practical use of bootstrap procedures in deep neural networks due to their high computational burden. To alleviate this, Shin et al. (2021) proposes Neural Bootstrapper (NeuBoots). NeuBoots circumvents multiple training of networks by learning a bootstrap generator.

**Random Weight Bootstrapping** Let $\sum_{c \in \mathcal{C}} \ell(f(x_c), y_c)$ be the loss function of interest for a neural network $f$. In standard bootstrap procedures, a bootstrapped neural network can be obtained by minimizing the loss function weighted by a random bootstrap weight $\mathbf{w}_\mathcal{C} := \{w_c : c \in \mathcal{C}\}$:

$$L(\mathbf{w}_\mathcal{C}, f, \mathcal{D}_\mathcal{C}) = \sum_{c \in \mathcal{C}} w_c \ell(f(x_c), y_c). \tag{5}$$

According to the choice of the distribution on $\mathbf{w}_\mathcal{C}$, various bootstrap procedures can be represented under the form of (5); e.g., the paired bootstrap by $\mathbf{w}_\mathcal{C} \sim \text{Multinomial}(n; 1/n, \ldots, 1/n)$ and the Random Weight Bootstrapping (RWB) (Præstgaard & Wellner, 1993; Newton & Raftery, 1994) by $\mathbf{w}_\mathcal{C} \sim |\mathcal{C}| \times \text{Dirichlet}(1, \ldots, 1)$. NeuBoots utilizes RWB to avoid the data discard problem which can occur in the standard bootstrapping. We then compute bootstrapped neural networks $\{\hat{f}^{(b)} : b = 1, \ldots, B\}$ via minimization of (5) for sampled $\mathbf{w}_\mathcal{C}^{(1)}, \ldots, \mathbf{w}_\mathcal{C}^{(B)}$.

**Learning To Generate Bootstrap Distribution**    The main idea of NeuBoots is to construct a single generative network that models the bootstrapped neural networks with varying bootstrap weights in (5). This formulation modifies the backbone network in a form of $f(x, \mathbf{w}_{\mathcal{C}})$ that inputs both feature $x$ and bootstrap weight $\mathbf{w}_{\mathcal{C}}$. Shin et al. (2021) show that the minimizer of the following loss generates valid bootstrap evaluations that match the results of the standard bootstrap procedure:

$$\mathcal{L}(f, \mathcal{D}_{\mathcal{C}}) = \mathbb{E}_{\mathbf{w}_{\mathcal{C}} \sim |\mathcal{C}| \times \text{Dirichlet}(1,\dots,1)} \left[ L(\mathbf{w}_{\mathcal{C}}, f(\cdot, \mathbf{w}_{\mathcal{C}}), \mathcal{D}_{\mathcal{C}}) \right],$$

We call this weighted bootstrapping loss. Once this generator is trained via a single optimization procedure, we can efficiently generate bootstrapped predictions by plugging random bootstrap weights in the trained generator; i.e., for a feature of interest $x$, the trained generator inputs $\{\mathbf{w}_{\mathcal{C}}^{(b)}\}_{b=1}^{B}$ and produces bootstrapped predictions $\hat{y}^{(b)} = \hat{f}(x, \mathbf{w}_{\mathcal{C}}^{(b)})$ for $b = 1, \dots, B$.

## 3    NEURAL BOOTSTRAPPING ATTENTIVE NEURAL PROCESSES

We propose a novel class of NP, called Neural Bootstrapping Attentive Neural Processes (NeuBANP). Aligned with the previous formulation of NP families, we can define our model $p_\theta$ as:

$$p_\theta(Y|X, \mathcal{D}_{\mathcal{C}}) = \int p_\varphi(Y|X, \mathbf{h}, \mathbf{z}) q(\mathbf{z}|\mathcal{D}_{\mathcal{C}}) \mathrm{d}\mathbf{z} = \int \prod_{i=1}^{n} p_\varphi(y_i|x_i, \mathbf{h}, \mathbf{z}) q(\mathbf{z}|\mathcal{D}_{\mathcal{C}}) \mathrm{d}\mathbf{z} \tag{6}$$

$$\approx \prod_{i=1}^{n} p_\varphi(y_i|x_i, \mathbf{h}, \mathbf{z}) \text{ where } \mathbf{z} \sim q(\mathbf{z}|\mathcal{D}_{\mathcal{C}}). \tag{7}$$

Here $p_\varphi$ is the decoder and $q$ denotes the posterior distribution. Since the above integral is intractable, we approximate the predictive distribution by sampling $\mathbf{z}$ from the bootstrap distribution $q(\cdot|\mathcal{D}_{\mathcal{C}})$, instead of Gaussian distribution as in (A)NP. Precisely, we train a generative encoder network $g_\phi$, which outputs bootstrapped representation pairs $(\mathbf{h}, \mathbf{z}) = \{(\mathbf{h}^{(b)}, \mathbf{z}^{(b)})\}_{b=1}^{B} = g_\phi(X, \mathcal{D}_{\mathcal{C}}, \{\mathbf{w}_{\mathcal{C}}^{(b)}\}_{b=1}^{B})$. Through the meta-learning framework (1), our model learns to generate bootstrapped predictions for an arbitrarily given function $f \sim \mathcal{P}$. Thus, our approach can be regarded as a *learn-to-bootstrap* method for stochastic processes. Compared to the fixed bootstrap method used in BANP, our learnable bootstrap method can find the best strategy to appropriately generate the random functions regarding the given context and the general property of the underlying data generating process. This will lead to the better modeling of functional uncertainty of the target stochastic processes. Also, note that NeuBANP can obtain a number of bootstrapped predictions by simply plugging different bootstrap weights into $g_\phi$, while BANP needs repetitive data resampling from scratch. Figure 2 shows the difference between the forward computation paths of BANP and NeuBANP in detail.

### 3.1    NEURAL BOOTSTRAPPING CONTEXTS WITH ATTENTION MODULES

To train a generative network, which outputs the bootstrapped representations, we modify the encoder in ANP to take both $(x, \mathcal{D}_{\mathcal{C}})$ and bootstrap weight $\mathbf{w}_{\mathcal{C}}$ as inputs. We introduce the *posterior* and the *prediction* paths in the encoder, analogous to NP's latent and deterministic paths. NeuBANP is designed to leverage RWB, which is theoretically proven as a valid bootstrap method. In detail, the posterior and prediction path in NeuBANP take the random bootstrap weight as an auxiliary input. This input provides enough randomness into the network so that our model can successfully capture the functional uncertainty.

**Posterior path**    We tag each context $(x_c, y_c) \in \mathcal{D}_{\mathcal{C}}$ with a bootstrap weight $w_c^{(b)}$ to construct bootstrapped contexts $\mathcal{D}_{\mathcal{C}}^{(b)} := \{(x_c, y_c, w_c^{(b)})\}_{c \in \mathcal{C}}$. Then the posterior path receives $\mathcal{D}_{\mathcal{C}}^{(b)}$ as input and outputs a latent variable $\mathbf{z}^{(b)}$. In detail, we apply self-attention to $\mathcal{D}_{\mathcal{C}}^{(b)}$ and multiply the resultant representation $\mathbf{z}_{\mathcal{C}}^{(b)}$ by the bootstrap weight $\mathbf{w}_{\mathcal{C}}^{(b)}$ before mean aggregation. This path connects the contexts with weighted bootstrapping loss during the training and allows the bootstrap weights to model bootstrapped posterior distribution $q$ by controlling the magnitude of each context representation.

**Prediction path**    The prediction path outputs a target-specific representation $\mathbf{h}^{(b)}$ that is relevant for the prediction. We apply self-attention to contexts $\mathcal{D}_{\mathcal{C}}$ and multiply the bootstrap weight $\mathbf{w}_{\mathcal{C}}^{(b)}$

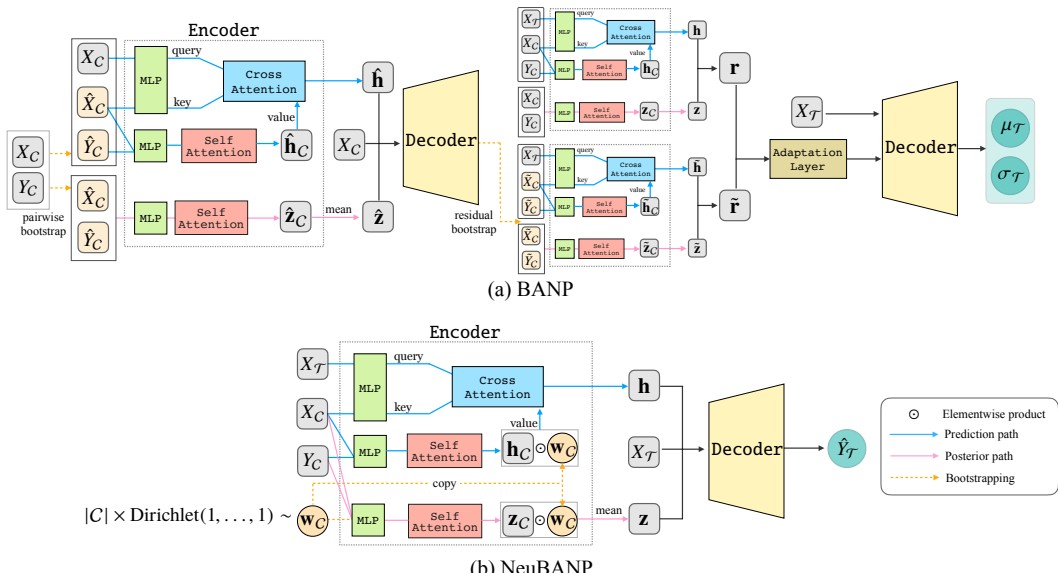

**Fig. 2.** A single forward computation of (a) BANP and (b) NeuBANP. Note that the inference for each target point requires $B$ times of this forward computation.

element-wisely to compute another bootstrapped representation. Cross-attention module uses $X_\mathcal{C}$ and this bootstrapped representation as *key-value* pairs to which the target *query* $x$ attends. Consequently, reflecting the bootstrap weights shared with the posterior path, the prediction path models interactions between the given context and the target input.

We summarize the above posterior and prediction paths as follows:

$$\mathbf{z}_\mathcal{C}^{(b)} = \{z_c^{(b)}\}_{c \in \mathcal{C}} = \text{SelfAttn}(\mathcal{D}_\mathcal{C}^{(b)}), \quad \mathbf{z}^{(b)} = \text{mean}(\mathbf{z}_\mathcal{C}^{(b)} \odot \mathbf{w}_\mathcal{C}^{(b)}) \tag{8}$$

$$\mathbf{h}_\mathcal{C} = \{h_c\}_{c \in \mathcal{C}} = \text{SelfAttn}(\mathcal{D}_\mathcal{C}), \quad \mathbf{h}^{(b)} = \text{CrossAttn}(x, X_\mathcal{C}, \mathbf{h}_\mathcal{C} \odot \mathbf{w}_\mathcal{C}^{(b)}) \tag{9}$$

where $\odot$ denotes element-wise multiplication. We concatenate random bootstrap weights to context data, unlike NeuBoots, which only utilizes random weight multiplication in the final layer. Concatenation of weights and the contexts in the posterior path yields two effects. First, concatenating random information in given contexts provides sufficient randomness to the model, allowing it to cover a wide range of function samples in a space where a true function is likely to exist. Second, we made this modification to maximize the use of random weights and provide bootstrapping information to the model without resampling the data, enabling better uncertainty estimation. Additionally, by multiplying the random weights to the representations from both paths, as in (8) and (9), the representations are consistent with the weights and maximize bootstrapping effect. We think that these multiplications also provide the effect of regularization, which mitigates the overfitting tendency which ANP and BANP show in a simple regression experiment (see Figure 1).

**Decoder and Bootsrapped Prediction** The decoder takes $(\mathbf{z}^{(b)}, \mathbf{h}^{(b)})$ from the encoder and target $x \in \mathcal{X}$ as inputs to generate a prediction $\hat{y}^{(b)} = \text{MLP}(x, \mathbf{z}^{(b)}, \mathbf{h}^{(b)})$. We can generate bootstrap samples $\hat{y}^{(1)}, \hat{y}^{(2)}, ..., \hat{y}^{(B)}$ by plugging $\mathbf{w}_\mathcal{C}^{(1)}, \mathbf{w}_\mathcal{C}^{(2)}, ..., \mathbf{w}_\mathcal{C}^{(B)}$ into the encoder $g_\phi(x, \mathcal{D}_\mathcal{C}, \cdot)$, respectively. We estimate the predictive mean and standard deviation using bootstrap samples:

$$\mu = \frac{1}{B}\sum_{b=1}^{B} \hat{y}^{(b)}, \quad \sigma = \sqrt{\frac{1}{B-1}\sum_{b=1}^{B}(\hat{y}^{(b)} - \mu)^2} \tag{10}$$

In the previous NP methods, the decoder directly outputs the parameters of the predictive distribution, but the decoder of NeuBANP outputs the stochastic predictions $\{\hat{y}^{(b)}\}_{b=1}^{B}$. This design of output is a distinction of our model from other NP methods and allows the nonparametric estimation. Due to this structure, existing NPs set the lower bound of standard deviation for robust performance.

We found that the performance was susceptible to the lower bound value. However, NEUBANP naturally obtains parameters of predictive distribution using bootstrap predictions and shows better performance without such heuristics. It also has the advantage of calculating higher-order statistics without changing the structure.

## 3.2 TRAINING

**Weighted Bootstrapping Loss**    We train NEUBANP with weighted loss similar to that of NeuBoots as demonstrated in Section 2.3. Only context data has the corresponding bootstrap weight in our setting, but the model still has to fit target data. Thus we designed loss function as a sum of weighted context loss $\mathcal{L}_{\text{context}}$ and non-weighted target loss $\mathcal{L}_{\text{target}}$ as follows:

$$\mathcal{L}_{\text{total}} = \underbrace{\frac{1}{B}\sum_{b=1}^{B}\frac{1}{|\mathcal{C}|}\sum_{c\in\mathcal{C}}\Big(-w_c^{(b)}\log\mathcal{N}(y_c|\mu_c,\sigma_c^2)\Big)}_{\mathcal{L}_{\text{context}}} + \underbrace{\frac{1}{|\mathcal{T}|}\sum_{t\in\mathcal{T}}\Big(-\log\mathcal{N}(y_t|\mu_t,\sigma_t^2)\Big)}_{\mathcal{L}_{\text{target}}} \quad (11)$$

where $\mu_c, \sigma_c, \mu_t$, and $\sigma_t$ are computed by (10) for context $(x_c, y_c) \in \mathcal{D}_{\mathcal{C}}$ and target $(x_t, y_t) \in \mathcal{D}_{\mathcal{T}}$. Averaging weighted context loss with multiple bootstrap weights improves the robustness of a model by showing various bootstrap samples during training. We trained the model with negative log-likelihood (NLL). One can replace the NLL with a different loss function, such as cross-entropy according to the target tasks.

## 4 EXPERIMENTS

We conducted experiments to compare NEUBANP with the previous NP methods for regression and sequential decision-making problems. For regression tasks, we conducted one-dimensional (1D) regression experiments on random functions generated from GP prior and image completion tasks as two-dimensional (2D) regression (see Appendix B.5 for image completion). For sequential decision-making problems, we evaluated each method in Bayesian optimization (BO) and Contextual Multi-Armed Bandit (CMAB).

### 4.1 NONPARAMETRIC REGRESSION AND UNCERTAINTY ESTIMATION

**Settings**    We followed the settings in Lee et al. (2020). To obtain meta-training datasets, we sampled batches of random functions from GP prior with RBF kernel, and context and target points were chosen randomly from each function. In addition, kernel parameters of GP were randomly sampled so that the models could learn about various functions. NEUBANP was trained with 10 bootstrap samples, and we confirmed that it is robust to the number of samples. Please refer to Appendix B.2 for details.

**Results**    The numerical results are summarized in Table 2. ANP and BANP tend to estimate homogeneous uncertainties for all target points in a situation where context points are sufficiently given (see Figure 3), because they place the heuristic lower bound on the variance. In the case of BANP, homoscedasticity occurs even when the number of context is small. Another problem only occurs in BANP, which is that it does not properly estimate functional uncertainty. Since the uncertainty about the shape of the true function is high in the region where the context point is not given, models should generate a wide range of function samples. However, we can see that BANP generates almost the same functions, not like those of the other methods, so we argue that BANP is not an appropriate method for modeling functional uncertainty. NEUBANP resolves these problems efficiently and achieves the best performance except for target prediction in the Periodic kernel.

### 4.2 BAYESIAN OPTIMIZATION

Since NP can approximate a class of arbitrary functions, it can replace GP, the commonly used surrogate model of BO. It is crucial to approximate the objective function from the given observations using the surrogate model, but evaluating the acquisition function and determining the subsequent samples are vital for efficient exploration. We evaluated the proposed method on various black-box functions, which may be unobserved in the meta-training step (see Algorithm 1).

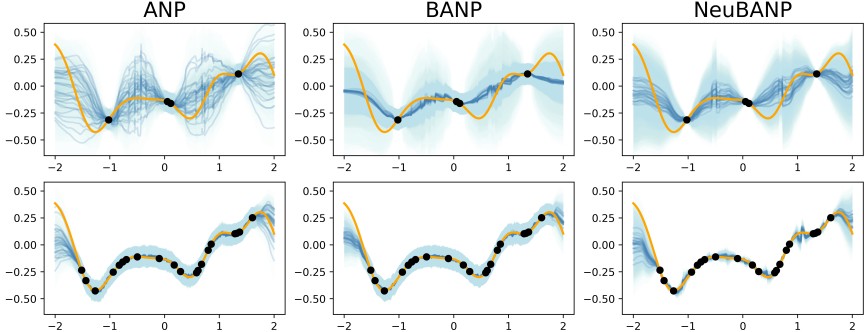

**Fig. 3.** Comparison of ANP, BANP, and NeuBANP in 1D regression given 4 context points (top) and 20 context points (bottom). Orange lines represent the ground-truth function. Blue lines are predictive mean given by each model and shaded region denotes the standard deviation (amount of uncertainty). To visualize the quality of functional uncertainty, we overlapped multiple shaded areas obtained with 30 sampled outputs for each input.

**Settings** For 1D, we followed the same setting in Lee et al. (2020). We set objective functions generated from GP with RBF, Matérn 5/2, and Periodic kernels and applied the models trained in Section 4.1. Furthermore, we demonstrated the BO performance of NeuBANP for multi-dimensional settings (2D and 3D). We set objective functions as various benchmark functions used in the optimization literature (Kim, 2020; Kim & Choi, 2017). See Appendix B.3 for details. A simple regret measured the performance of each model, and the mean performance over 100 experiments is reported for reliable evaluations.

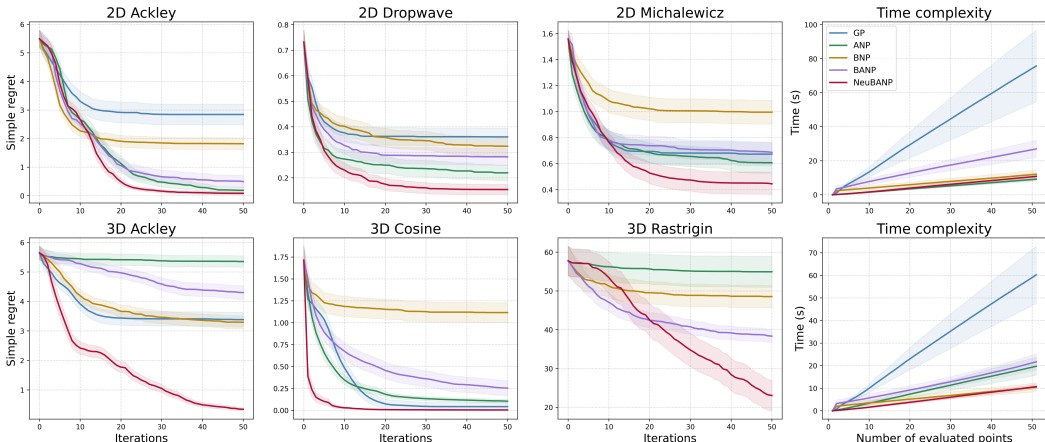

**Fig. 4. Left:** Multi-dimensional Bayesian optimization results on various benchmark functions with UCB as an acquisition function. Bold lines represent the mean performance over 100 experiments. We indicate 20% of the standard deviation. **Right most:** Time complexity of each model as the number of observations increases.

**Results** NeuBANP outperformed the other methods in BO experiments (see Figure 4 and 8). Table 3 and 4 shows numerical results. For multi-dimensional BO, NeuBANP achieved the best results for every function except the Hartmann-3D when using Upper Confidence Bound (UCB) as an acquisition function. For the Hartmann-3D, the performance of NeuBANP is statistically comparable to the best method (ANP) when considering the standard deviation. For Goldsteinprice and Rastrigin functions, GP records poor performance due to its numerical errors during the optimization procedure (see Appendix B.3). The rightmost plots of Figure 4 show the time complexity according to the number of observation points. NeuBANP has the fastest decreasing rate of regrets in terms of iterations in both cases. We also conducted BO experiments using Expected Improvement (EI) to test the performance of NeuBANP to be independent of the selection of acquisition functions (Figure 9). Figure 10 demonstrates the better exploration strategy of NeuBANP compared to BANP. The second

| Regret | Method | $\delta = 0.5$ | $\delta = 0.7$ | $\delta = 0.9$ | $\delta = 0.95$ | $\delta = 0.99$ |
|---|---|---|---|---|---|---|
| | Uniform | $100.00 \pm 0.08$ | $100.00 \pm 0.09$ | $100.00 \pm 0.25$ | $100.00 \pm 0.37$ | $100.00 \pm 0.78$ |
| | Neural Linear | $0.95 \pm 0.02$ | $1.60 \pm 0.03$ | $4.65 \pm 0.18$ | $9.56 \pm 0.36$ | $49.63 \pm 2.41$ |
| | MAML | $2.95 \pm 0.12$ | $3.11 \pm 0.16$ | $4.84 \pm 0.22$ | $7.01 \pm 0.33$ | $22.93 \pm 1.57$ |
| Cumulative | NP | $1.60 \pm 0.06$ | $1.75 \pm 0.05$ | $3.31 \pm 0.10$ | $5.71 \pm 0.24$ | $22.13 \pm 1.23$ |
| | ANP | $2.17 \pm 1.89$ | $3.59 \pm 8.03$ | $5.63 \pm 8.48$ | $11.68 \pm 8.97$ | $24.75 \pm 7.08$ |
| | BANP | $2.04 \pm 1.52$ | $2.34 \pm 1.23$ | $4.30 \pm 0.77$ | $6.76 \pm 1.03$ | $21.18 \pm 1.69$ |
| | NeuBANP | $\mathbf{0.85} \pm 0.22$ | $\mathbf{1.02} \pm 0.27$ | $\mathbf{1.85} \pm 0.56$ | $\mathbf{3.04} \pm 0.88$ | $\mathbf{9.76} \pm 1.93$ |
| | Uniform | $100.00 \pm 0.45$ | $100.00 \pm 0.78$ | $100.00 \pm 1.18$ | $100.00 \pm 2.21$ | $100.00 \pm 4.21$ |
| | Neural Linear | $\mathbf{0.33} \pm 0.04$ | $\mathbf{0.79} \pm 0.07$ | $2.17 \pm 0.14$ | $4.08 \pm 0.20$ | $35.89 \pm 2.98$ |
| | MAML | $2.49 \pm 0.12$ | $3.00 \pm 0.35$ | $4.75 \pm 0.48$ | $7.10 \pm 0.77$ | $22.89 \pm 1.41$ |
| Simple | NP | $1.04 \pm 0.06$ | $1.26 \pm 0.21$ | $2.90 \pm 0.35$ | $5.45 \pm 0.47$ | $21.45 \pm 1.3$ |
| | ANP | $0.99 \pm 1.68$ | $1.50 \pm 2.21$ | $3.64 \pm 4.71$ | $6.32 \pm 7.33$ | $21.65 \pm 1.72$ |
| | BANP | $1.22 \pm 1.83$ | $2.37 \pm 3.04$ | $3.27 \pm 5.33$ | $7.73 \pm 12.16$ | $20.63 \pm 34.21$ |
| | NeuBANP | $0.86 \pm 0.06$ | $1.04 \pm 0.08$ | $\mathbf{1.88} \pm 0.14$ | $\mathbf{3.09} \pm 0.23$ | $\mathbf{9.96} \pm 0.70$ |

**Table 1.** Results of the wheel bandit problem according to the value of $\delta$. Mean and standard deviation for cumulative regret and simple regret over 50 runs are reported. Regrets are normalized to that of the uniform policy.

and fourth columns show the explored points by BANP and NeuBANP, respectively. In most cases, NeuBANP converged to the optimum faster than BANP. The third and fifth column shows the contour plots of acquisition functions of each method. Note that NeuBANP accurately infers the potential area of the optimum compared to BANP. As explained above, NeuBANP can estimate heterogeneous uncertainty, and thus it is able to handle well for the exploration-exploitation trade-off, which is essential in sequential decision-making problems.

## 4.3 CONTEXTUAL MULTI-ARMED BANDIT

We conducted a CMAB experiment, the *wheel bandit* problem, as in Garnelo et al. (2018b) to show that NeuBANP works efficiently as well as BO based on its performance of uncertainty estimation.

**Settings**   We followed the same environment in Garnelo et al. (2018b), but used UCB policy. For more details, please refer to Appendix B.4. The parameter $\delta$ determines the environment of the wheel bandit problem. As $\delta$ increases, high-reward observation becomes sparse, which makes the problem more difficult. We set the baselines of CMAB experiment to MAML (Finn et al., 2017), Neural Linear (Riquelme et al., 2018) and NP. We measured cumulative regrets and simple regrets for 2,000 iterations to demonstrate the performance of each model.

**Results**   NeuBANP performed well for various $\delta$ values (see Table 1). Note that NeuBANP showed better performance in challenging environments with sparse high rewards. NeuBANP showed the ability to learn various reward distributions based on appropriate functional uncertainty modeling. The result demonstrates that NeuBANP can utilize a small number of context information and make accurate estimations. NeuBANP also has stable results as its small variance shows. On the other hand, ANP has extremely high variance in the performance because it overfits to certain prediction and failed to adapt to the various bandit environments.

## 5 RELATED WORK

**Neural Processes**   CNP (Garnelo et al., 2018a) uses a pair of encoder and decoder networks to produce a predictive posterior distribution over functions. NP (Garnelo et al., 2018b) introduces a global latent variable to embed functional uncertainty in the deterministic architecture of CNP and predicts various outputs given the same context data. ANP (Kim et al., 2018) then enhances predictive accuracy by replacing MLP modules in NP with the attention modules. BNP (Lee et al., 2020) proposes the bootstrap method to model uncertainty in stochastic processes without the assumption of a single latent variable on which previous methods rely. In addition to these works, there are many attempts to use NPs in various tasks. Singh et al. (2019) tackles sequential stochastic processes, where the

dynamics of the given system changes as the time being. Leveraging time-variant context points, Singh et al. (2019) models the underlying temporal 3D structures. Gordon et al. (2020) extends NP families to contain translation equivariant functions, providing theoretical formulation to represent translation-invariant functional representations. Louizos et al. (2019) do not assume explicit global latent variables, instead supported by dependency graph among local latent variables, to encode inductive bias for given data easier than NPs that use global latent variables.

**Bootstrapping Neural Networks**   Bootstrap method (Efron, 1987) is a reliable approach to estimate predictive uncertainty (Lakshminarayanan et al., 2017; Osband et al., 2016). However, it is computationally inefficient to go through the feed-forward computation as much as the number of bootstraps; hence, it discourages the practical application of bootstrap in neural networks. There have been several works to circumvent this issue by approximating bootstrapped distribution. Amortized bootstrap (Nalisnick & Smyth, 2017) approximates bootstrap distribution over model parameters by using amortized inference and implicit models. Generative Bootstrap Sampler (Shin et al., 2020) proposes a computational bootstrap procedure that constructs a generator function of bootstrap evaluations for classical statistical models. Neural Bootstrapper (Shin et al., 2021) suggests a simple recipe for generating bootstrapped predictive distributions of MLPs and convolutional neural networks.

**Meta-Learning based Stochastic Optimization**   NPs are trained with a meta-learning framework to solve various tasks related to the data generation process through a single optimization. Santoro et al. (2016); Chen et al. (2017) follow the same training procedure of NP. However, Santoro et al. (2016) proposes an memory-augmented network for robust meta-learning, while Chen et al. (2017) proposes the method to produce an algorithm for black-box optimization using recurrent networks. Sharaf & Daumé III (2019) presents a meta-learning algorithm for learning a good exploration policy in the contextual bandit. Ravi & Beatson (2019) also solves contextual bandits based on the Bayesian framework by inferring a posterior on weights of neural networks. Galashov et al. (2019) introduces a unified framework for applying NP to a wide range of sequential decision-making problems.

## 6   Conclusion

We have proposed NeuBANP, a novel bootstrap method for a family of NP to model functional uncertainty appropriately. Instead of the standard bootstrap, NeuBANP learns to construct a generator function that produces valid bootstrapped distribution without resampling which can be considered as a *learn-to-bootstrap* method. NeuBANP successfully worked in a meta-learning framework, providing diverse trajectories of underlying data-generating processes, consistent to any given context. In addition, NeuBANP estimates the local uncertainty accurately, resolving overfitted prediction and variance overestimation problems observed in both ANP and BANP. We replace the additional layer and repetitive computations in BANP with the simple attachment of bootstrap weights to the model, which leads to lower computations and smaller memory. NeuBANP shows superior performance to previous NP methods in regression and stochastic optimization tasks, including multi-dimensional setting, which has been the desired application of NP. However, due to the numerical instability of GP, there is a limit in sampling high-dimensional stochastic processes for the meta-learning framework. We suggest that this is a primary task for scalable applications of NP in stochastic optimization, as a challenging research direction.

## 7   Reproducibility

For reproducibility of experimental results, we provide a link to anonymous github that contains our source code in Appendix B. The source code includes the implementation of our model, data generation, and experiments. Additionally, the data generation steps are thoroughly explained in Appendix B.

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

## A  MODEL

### A.1  BASIC OPERATION

**Muti-Layer Perceptron**  $\text{MLP}(d_i, d_h, d_o, n_l)$ denotes the multi-layer perceptron consisting of $n_l$ linear transformations and ReLU activations between them. Let parameters $d_i, d_h, d_o$ the dimension of input, hidden and output feature, repectively. We used same $d_h$ for every hidden layers. $\text{Lin}(p, q)$ denotes the linear transformation of input with feature dimension $p$ into output with feature dimension $q$.

$$\text{MLP}(d_i, d_h, d_o, n_l)(X) = \text{Lin}(d_h, d_o) \circ (\text{ReLU} \circ \text{Lin}(d_h, d_h))^{n_l - 2} \circ \text{ReLU} \circ \text{Lin}(d_i, d_h)(X) \tag{12}$$

**Dot Product Attention**  DotProdAttn (Vaswani et al., 2017) denotes the attention operation with attention score based on cosine similarity. Dot product of query (Q) and key (K) tensors calculates the similarity of each vectors in query tensor relative to each vectors in key tensor. Attention score is calculated through softmax operation of normalized similiarity. Let $d_k$ the feature dimension of key and query tensors, and $d_v$ that of value tensors.

$$\text{DotProdAttn}(Q, K, V) = \text{softmax}\big(Q^T K / \sqrt{d_k}\big) V \tag{13}$$

**Multi-Head Attention** MHA denotes multi-head attention with dot product attention. The input is pre-processed with MLP and the output of attention is post-processed with layer normalization. For simplicity, we omit parameters of each MLP layers. $\text{split}(X, n)$ denotes splitting of tensor $X$ with respect to feature axis into a tuple of $n$ tensors $(X_i')_{i=1}^n$ which have same dimension in feature axis. $[X_1, X_2, ..., X_n]$ denotes the concatenation of tensors $X_1, X_2, ..., X_n$ with respect to feature axis. LayerNorm denotes the layer normalization introduced in Ba et al. (2016). Let $n_{head}$ the number of heads in multi-head attention.

$$(Q_i')_{i=1}^{n_{head}} = \text{split}(\text{MLP}_{qk}(Q), n_{head}) \tag{14}$$

$$(K_i')_{i=1}^{n_{head}} = \text{split}(\text{MLP}_{qk}(K), n_{head}) \tag{15}$$

$$(V_i')_{i=1}^{n_{head}} = \text{split}(\text{MLP}_v(V), n_{head}) \tag{16}$$

$$\text{MHA}(Q, K, V) = \text{LayerNorm}([(\text{DotProdAttn}(Q_i', K_i', V_i'))_{i=1}^{n_{head}}]) \tag{17}$$

**Self-Attention** We used self-attention to calculate efficient representations of context $\mathcal{D}_{\mathcal{C}} = (X_{\mathcal{C}}, Y_{\mathcal{C}})$. We define self-attention based on multi-head attention as follows:

$$\mathcal{D}_{\mathcal{C}}' = [X_{\mathcal{C}}, Y_{\mathcal{C}}] = ([x_c, y_c])_{c \in \mathcal{C}}, \quad \text{SelfAttn}(\mathcal{D}_{\mathcal{C}}) := \text{MHA}(\mathcal{D}_{\mathcal{C}}', \mathcal{D}_{\mathcal{C}}', \mathcal{D}_{\mathcal{C}}') \tag{18}$$

**Cross-Attention** We used cross-attention to calculate representation of context specific to target feature $x$ in interest, when original representations $(h_c)_{c \in \mathcal{C}}$ is given. We define cross-attention based on multi-head attention as follows:

$$\text{CrossAttn}(x, X_{\mathcal{C}}, (h_c)_{c \in \mathcal{C}}) := \text{MHA}(x, X_{\mathcal{C}}, [(h_c)_{c \in \mathcal{C}}]). \tag{19}$$

## A.2 ARCHITECTURE

For fair comparison, we used the same architecture of all models as in Lee et al. (2020). For NEUBANP, we increased the input dimension of SelfAttn in the posterior path by one to take bootstrap weight as additional input. Please refer to Lee et al. (2020) for detailed model architecture of ANP and BANP.

## A.3 NON-ATTENTIVE CASE

As an ablation study, we consider the non-attentive case of NEUBANP, called Neural Bootstrapping Neural Processes (NEUBNP). Like the architecture of NEUBANP is based on ANP, the architecture of this model is based on NP, the non-attentive counterpart of ANP. With $\mathcal{D}_{\mathcal{C}}^{(b)} = \{(x_c, y_c, w_c^{(b)})\}_{c \in \mathcal{C}}$ as defined in 3, we applied the similar strategy of using random weights in the encoder as follows, and trained with the same loss function:

$$\mathbf{z}_{\mathcal{C}}^{(b)} = \{z_c^{(b)}\}_{c \in \mathcal{C}} = \text{MLP}(\mathcal{D}_{\mathcal{C}}^{(b)}), \quad \mathbf{z}^{(b)} = \text{mean}(\mathbf{z}_{\mathcal{C}}^{(b)} \odot \mathbf{w}_{\mathcal{C}}^{(b)}) \tag{20}$$

$$\mathbf{h}_{\mathcal{C}}^{(b)} = \{h_c^{(b)}\}_{c \in \mathcal{C}} = \text{MLP}(\mathcal{D}_{\mathcal{C}}^{(b)}), \quad \mathbf{h}^{(b)} = \text{mean}(\mathbf{h}_{\mathcal{C}}^{(b)} \odot \mathbf{w}_{\mathcal{C}}^{(b)}) \tag{21}$$

$$\tag{22}$$

To make the non-attentive counterpart of NEUBANP, we use random weights in both paths, resulting in two latent variables $z^{(b)}, h^{(b)}$. They induce randomness into the decoder output $\hat{y}^{(b)} = \text{MLP}(x, \mathbf{z}^{(b)}, \mathbf{h}^{(b)})$, and the predictive distribution is construct by (10). The results in Table 2 and Table 3 shows that this model performs worse than NEUBANP, but outperforms BNP and NP, showing the quality of functional uncertainty the model learns with the bootstrap.

## B EXPERIMENTS

Implementation[2] of NPs other than NEUBANP was borrowed from the source code of Lee et al. (2020)[3]. Regression and Bayesian optimization experiment was done in single *GeForce RTX 2080 Ti* GPU with the memory of $11,019$ MiB. Multi-dimensional regression including image completion was done in *Tesla V100* GPU with the memory of $32,480$ MiB.

---

[2] https://anonymous.4open.science/r/neubanp_initial
[3] https://github.com/juho-lee/bnp, MIT License.

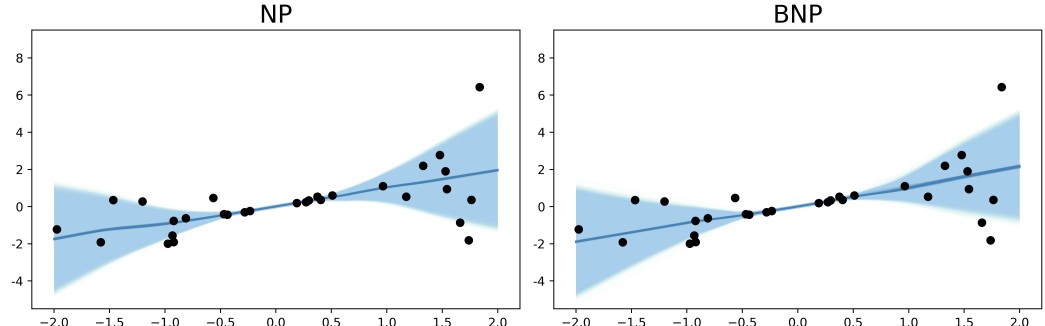

**Fig. 5.** Each plot shows predictions given by NP, BNP in a linear regression. The ground-truth function is a simple linear function with heterogeneous variance: $y = x + \beta\epsilon(x)$ where $\epsilon(x) \sim N(0, \sigma^2(x))$ and $\sigma(x) = \sqrt{x^2 + 10^{-5}}$. We used the official code provided by Lee et al. (2020). Unlike the case of ANP and BANP, considering that overfitting does not occur in NP and BNP, we can notice that attention modules are the leading cause of overfitting.

### B.1 SIMPLE LINEAR REGRESSION

We experimented using a linear function rather than a function sampled from the GP to analyze how well the NP models work in a simple regression task. Additionally, we need to examine how well the NP models predict uncertainty in the presence of heterogeneous noise in the data, and we add $\epsilon(x) \sim \mathcal{N}(0, \sigma^2(x)), \sigma(x) = \sqrt{x^2 + 10^{-5}}$ to the linear function. We multiplied the noise by coefficient $\beta$ to follow the meta-learning framework, and the $\beta$ value was uniformly randomly sampled from $[0.1, 1.0]$ during training. In this task, NP, BNP, and NeuBANP were able to estimate the underlying true linear function and the uncertainty of heterogeneous noise. On the other hand, in case of ANP and BANP, overfitting occurred and failed to predict the true function and its uncertainty (see Figures 1 and 5). As explained in the main text, we can see that the overfitting phenomenon occurs because of the attention mechanism that has appeared to solve the underfitting issue of the NP. NeuBANP has the advantage of the attention mechanism and achieved better performance through regularization using random weights. For (A)NP and B(A)NP, we used the official code provided by Lee et al. (2020). The remaining training settings are the same as the 1D regression in Appendix B.2, and we changed only the training iteration to $10,000$.

### B.2 1D REGRESSION

**Training** For all models, training dataset consists of randomly sampled context and target from functions following GP with RBF kernel $k(x, y) = s^2 \cdot \exp(-||x - y||^2/(2l^2))$. Parameters of kernel are randomly sampled with $s \sim \text{Uniform}(0.1, 1.0)$ and $l \sim \text{Uniform}(0.1, 0.6)$. Feature values $(x_i)_{i \in \mathcal{C} \cup \mathcal{T}}$ is chosen uniformly at random in $[-2, 2]$. The size of context and target are randomly sampled with $|\mathcal{C}| \sim \text{Uniform}(3, 47)$ and $|\mathcal{T}| \sim \text{Uniform}(3, 50 - |\mathcal{C}|)$. We trained all models for 100,000 iterations and used Adam optimizer (Kingma & Ba, 2015). For stable learning, we used the cosine annealing scheduler with initial learning rate $5 \times 10^{-4}$.

**Results** Table 2 shows log-likelihood of NPs for various evaluation dataset sampled from GP with RBF, Matérn 5/2, Periodic kernel. As in generation of training dataset, parameters of Matérn 5/2 kernel $k(x, y) = s^2(1 + \sqrt{5}||x - y||^2/(3l^2))\exp(-\sqrt{5}||x - y||/l)$ and Periodic kernel $k(x, y) = s^2 \exp(-2\sin^2(\pi||x - y||^2/p)/l^2)$ was randomly sampled with $s \sim \text{Uniform}(0.1, 1.0)$, $l \sim \text{Uniform}(0.1, 0.6)$ and $p \sim \text{Uniform}(0.1, 0.5)$. For RBF and Matérn 5/2 dataset, NeuBANP showed state-of-the-art performance both in fitting context and predicting target. We added figures showing the predictions of ANP, BANP, and NeuBANP for the Matérn 5/2 and Periodic kernel (see Figure 6 and 7). In the case of the Matérn kernel, the two problems described in the main text occurred identically for ANP and BANP (see Section 4.1). However, in the case of Periodic, we can see that all models failed to approximate the true function correctly. This result came out because we experimented with testing the models' generalization performance when trained with the RBF kernel. And the quantitative results in Table 2 show that the prediction performance of the attention-based

| Method | RBF | | Matérn 5/2 | | Periodic | |
|---|---|---|---|---|---|---|
| | context | target | context | target | context | target |
| CNP | $1.17 \pm 0.08$ | $0.87 \pm 0.36$ | $1.06 \pm 0.11$ | $0.65 \pm 0.39$ | $-0.31 \pm 0.41$ | $-2.05 \pm 1.17$ |
| NP | $1.11 \pm 0.09$ | $0.78 \pm 1.47$ | $0.99 \pm 0.11$ | $0.56 \pm 0.50$ | $-0.28 \pm 0.37$ | $-1.73 \pm 1.09$ |
| ANP | $1.38 \pm 0.00$ | $1.08 \pm 0.41$ | $1.38 \pm 0.00$ | $0.94 \pm 0.47$ | $0.21 \pm 0.76$ | $-6.82 \pm 2.83$ |
| BNP | $1.20 \pm 0.07$ | $0.92 \pm 0.34$ | $1.09 \pm 0.09$ | $0.72 \pm 0.35$ | $-0.18 \pm 037$ | $\mathbf{-1.16} \pm 0.56$ |
| BANP | $1.38 \pm 0.00$ | $1.12 \pm 0.33$ | $1.38 \pm 0.00$ | $0.99 \pm 0.38$ | $0.28 \pm 0.69$ | $-5.69 \pm 2.37$ |
| NeuBNP | $1.54 \pm 0.20$ | $1.01 \pm 0.55$ | $1.22 \pm 0.22$ | $0.53 \pm 0.59$ | $-0.34 \pm 0.45$ | $-2.34 \pm 1.95$ |
| NeuBANP | $\mathbf{3.17} \pm 0.28$ | $\mathbf{1.38} \pm 0.60$ | $\mathbf{3.09} \pm 0.29$ | $\mathbf{1.13} \pm 0.64$ | $\mathbf{1.56} \pm 0.73$ | $-11.49 \pm 8.08$ |

**Table 2.** Log-Likelihood of NPs for 48,000 different evaluations of context and target.

model on the target data is poor. Among them, NeuBANP has the worst performance, and we think the reason is the lower bound on the predicted variance set by ANP and BANP. Quantitative results show similar predictions for all models, but ANP and BANP achieve numerically more robust performance by setting a lower bound on the variance. We find some cases with the jumps in the function values which do not seem like a smooth function (See Figure 3, 6, and 7). This phenomena occurs in every attentive models including ANP, BANP, and NeuBANP. It looks like a distorted prediction on particular region, however, since our model predicts high variance in such region, this does not raise a problem in predicting the global trend, as we can see in high average log likelihood in 1d regression.

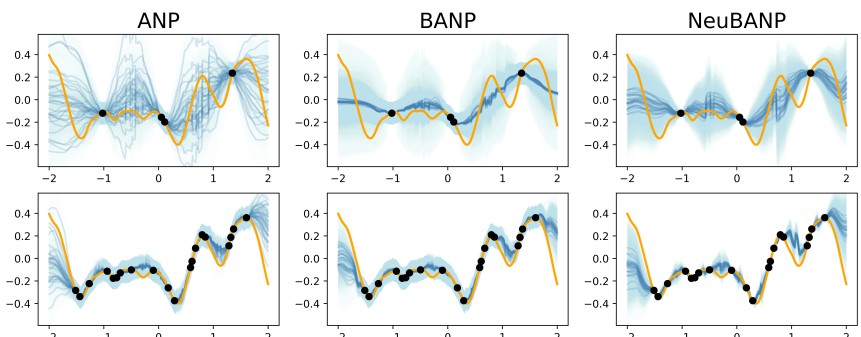

**Fig. 6.** Comparison of ANP, BANP, and NeuBANP in 1D regression. Matérn 5/2 kernel case.

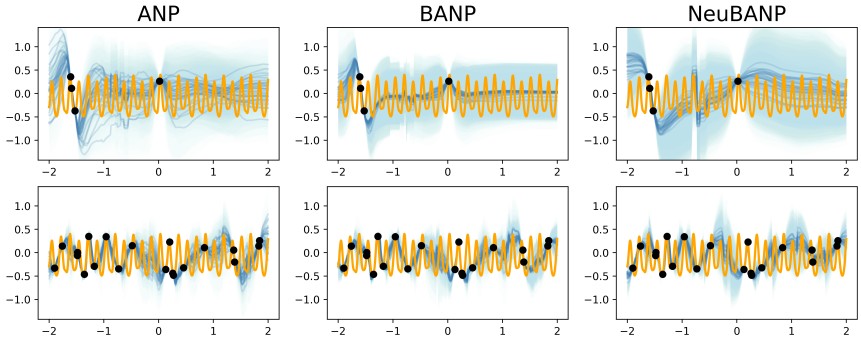

**Fig. 7.** Comparison of ANP, BANP, and NeuBANP in 1D regression. Periodic kernel case.

### B.3 BAYESIAN OPTIMIZATION

**One-dimensional case** Table 3 shows the performance of GP and various NPs for 1D Bayesian optimization task. RBF, Matérn 5/2, and Periodic kernels are used to generate evaluation dataset.

---

**Algorithm 1:** Neural Process based Bayesian Optimization.

**Input** : Target function $f^\star$; Acquisition function $\mathcal{U}$; Observed data $\mathcal{D}_0 = \{(x_0, f^\star(x_0))\}$; Maximum evaluation step $T$.

1 Meta-train a neural process $p_\theta$ on $f \sim \mathcal{P}(\mathcal{F})$.

2 **for** $t = 1, \ldots, T$ **do**

3      Find $x_t$ by optimizing acquisition function: $x_t = \underset{x \in \mathcal{X}}{\arg\min}\, \mathcal{U}\left(p_\theta(y|x, \mathcal{D}_{t-1})\right)$

4      Evaluate $f^\star(x_t)$ and update the observed data: $\mathcal{D}_t \leftarrow \mathcal{D}_{t-1} \cup \{(x_t, f^\star(x_t))\}$

---

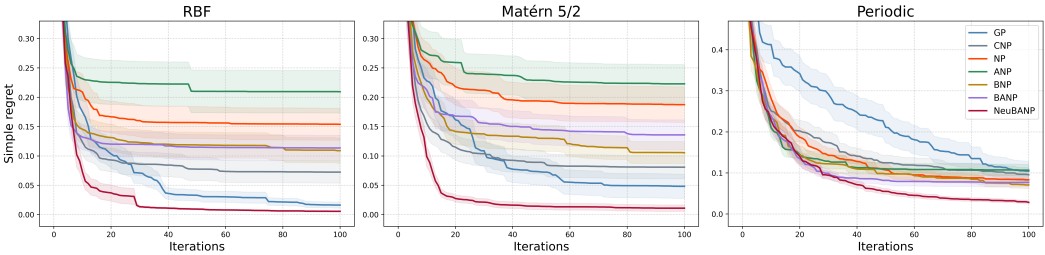

**Fig. 8.** 1D Bayesian optimization results. Bold lines show the mean of simple regrets over 100 experiments. We also report 10% of the standard deviation.

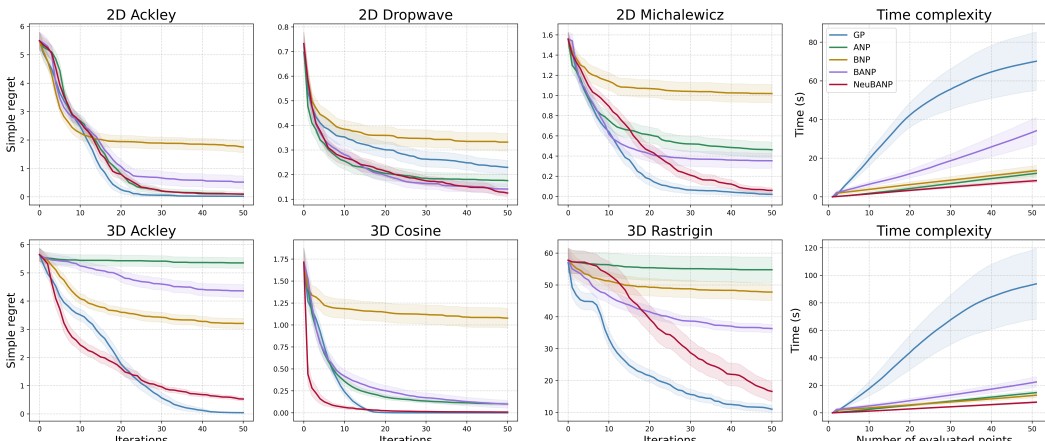

**Fig. 9. Left:** Multi-dimensional Bayesian optimization results on various benchmark functions with EI as an acquisition function. Bold lines represent the mean performance over 100 experiments. We indicate 20% of the standard deviation. **Right most:** Time complexity of each model as the number of observations increases.

**Multi-dimensional case** In multi-dimensional BO experiments, we used GPyTorch [4] (Gardner et al., 2018) for scalable GP regression, and BoTorch [5] (Balandat et al., 2020) for overall BO process (e.g., optimization of acquisition functions). GP was set to the default setting of BoTorch. In detail, GP model was parameterized with Matérn 5/2 kernel with ARD and constant mean function, and prior distribution for hyperparameters was set as Gamma$(3, 6)$ for length scale $l$ and Gamma$(2, 0.15)$ for output scale $s$. For three-dimensional BO experiment in Figure 4, the overall time complexity of ANP and BANP is almost the same. This result is seemingly in contrast with the fact that ANP takes a shorter time for prediction than BANP. However, since the BO algorithm contains the optimization of the acquisition function, the qualities of acquisition functions obtained by the model predictions may affect the overall time complexity. We conjecture that BANP gives an acquisition function easier to optimize than that of ANP so that the overall time complexities of both models are similar. Additionally, we did not report the results of GP for the two functions. Specifically, we omitted the Goldstein-Price

---

[4] https://github.com/cornellius-gp/gpytorch, MIT License.
[5] https://github.com/pytorch/botorch, MIT License.

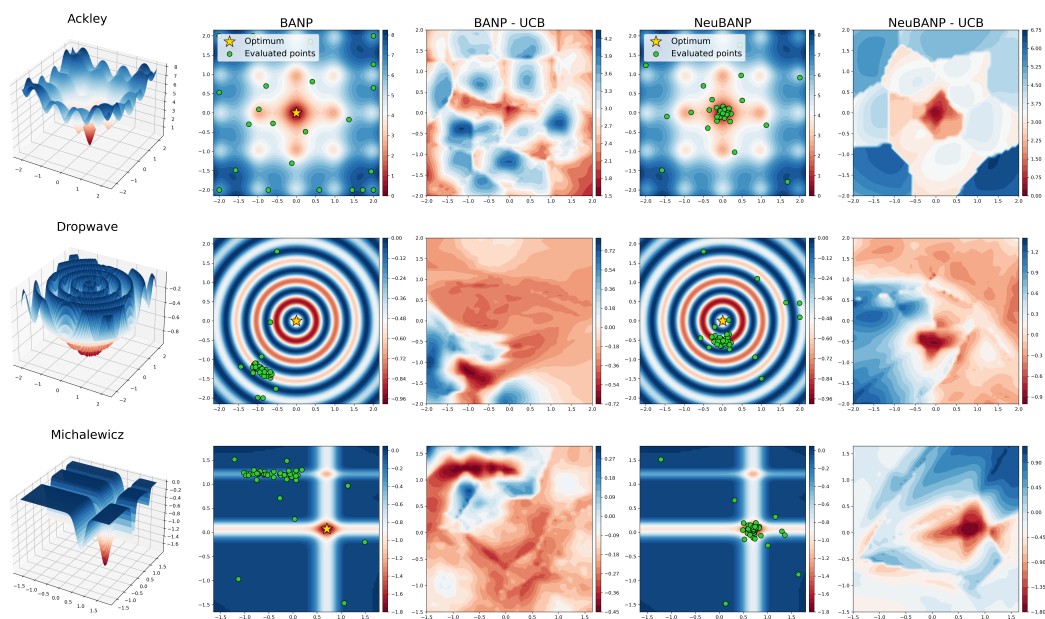

**Fig. 10. First column:** 2D objective functions **Second & Fourth columns:** Contour plots of functions and the evaluated points during Bayesian optimization. **Third & Fifth columns:** UCB value at the last iteration.

| Method | RBF | Matérn 5/2 | Periodic |
|---|---|---|---|
| GP (RBF) | $0.016 \pm 0.052$ | $0.048 \pm 0.206$ | $0.104 \pm 0.242$ |
| CNP | $0.072 \pm 0.188$ | $0.081 \pm 0.198$ | $0.096 \pm 0.166$ |
| NP | $0.154 \pm 0.273$ | $0.187 \pm 0.303$ | $0.083 \pm 0.121$ |
| ANP | $0.209 \pm 0.364$ | $0.223 \pm 0.328$ | $0.107 \pm 0.142$ |
| BNP | $0.109 \pm 0.214$ | $0.105 \pm 0.188$ | $0.071 \pm 0.091$ |
| BANP | $0.114 \pm 0.216$ | $0.136 \pm 0.256$ | $0.077 \pm 0.11$ |
| NeuBNP | $0.069 \pm 0.169$ | $0.125 \pm 0.238$ | $0.058 \pm 0.069$ |
| NeuBANP | $\mathbf{0.006} \pm 0.011$ | $\mathbf{0.011} \pm 0.055$ | $\mathbf{0.028} \pm 0.035$ |

**Table 3.** 1D Bayesian optimization results. Mean and standard deviations of simple regrets over 100 runs are reported.

since the simple regret value of GP was too large (the performance was poor) compared to other models and omitted the Rastrigin since the 46 errors occurred out of 100 experiments. An error may occur when ill-conditioned data is given during the kernel training process of the GP, and the data recommended by the UCB during the exploration (or exploitation) process seems to correspond to this condition. If we report the performance ignoring the numerical error, the simple regret for the Goldstein-Price was 52049.13, and the simple regret for the Rastrigin was 18.48. For the Rastrigin function, GP with UCB is numerically unstable, but like EI, it achieved the best performance.

## B.4 CONTEXTUAL MULTI-ARMED BANDIT

**Setting** We followed wheel bandit setting introduced in Riquelme et al. (2018). At every step $t\,(<T)$, a two-dimensional point $(x_t, y_t)$ inside the unit circle $R_{\mathtt{unit}} = \{(x, y) \in \mathbb{R}^2 : x^2 + y^2 \leq 1\}$ is given as a context $c_t$. The algorithm chooses an action $a_t \in \{1, 2, 3, 4, 5\}$. The stochastic reward $r_t = r(a_t, c_t)$ is sampled from the reward distribution. Let $R_1, R_2, R_3, R_4, R_5 \subset R_{\mathtt{unit}}$ the disjoint

| Dim | Target | GP | ANP | BNP | BANP | NeuBANP |
|---|---|---|---|---|---|---|
| 2D | Ackley | $2.84 \pm 1.82$ | $0.19 \pm 0.53$ | $1.82 \pm 1.03$ | $0.50 \pm 1.03$ | $\mathbf{0.08} \pm 0.27$ |
| | Dropwave | $0.36 \pm 0.17$ | $0.22 \pm 0.15$ | $0.32 \pm 0.18$ | $0.28 \pm 0.17$ | $\mathbf{0.15} \pm 0.10$ |
| | Goldsteinprice | - | $475.52 \pm 469.15$ | $2098.22 \pm 1509.88$ | $80.67 \pm 65.85$ | $\mathbf{30.33} \pm 25.42$ |
| | Michalewicz | $0.67 \pm 0.45$ | $0.61 \pm 0.40$ | $1.00 \pm 0.46$ | $0.69 \pm 0.38$ | $\mathbf{0.45} \pm 0.42$ |
| 3D | Ackley | $3.39 \pm 1.25$ | $5.36 \pm 0.97$ | $3.29 \pm 1.11$ | $4.30 \pm 1.15$ | $\mathbf{0.34} \pm 0.26$ |
| | Cosine | $0.04 \pm 0.24$ | $0.10 \pm 0.10$ | $1.12 \pm 0.57$ | $0.25 \pm 0.43$ | $\mathbf{0.005} \pm 0.003$ |
| | Hartmann | $0.42 \pm 0.77$ | $\mathbf{0.33} \pm 0.50$ | $1.94 \pm 0.86$ | $0.93 \pm 0.98$ | $0.39 \pm 0.39$ |
| | Rastrigin | - | $54.94 \pm 19.84$ | $48.55 \pm 14.34$ | $38.36 \pm 8.20$ | $\mathbf{23.06} \pm 19.77$ |

**Table 4.** Multi-dimensional Bayesian optimization results. Mean and standard deviations of simple regrets over 100 runs are reported.

sets (regions) of unit circle as follows:

$$R_1 = \{(x,y) : x^2 + y^2 < \delta\} \tag{23}$$

$$R_2 = \{(x,y) : \delta \le x^2 + y^2 \le 1, x > 0, y > 0\} \tag{24}$$

$$R_3 = \{(x,y) : \delta \le x^2 + y^2 \le 1, x < 0, y > 0\} \tag{25}$$

$$R_4 = \{(x,y) : \delta \le x^2 + y^2 \le 1, x < 0, y < 0\} \tag{26}$$

$$R_5 = \{(x,y) : \delta \le x^2 + y^2 \le 1, x > 0, y < 0\} \tag{27}$$

where the constant $\delta$ determines the size of $R_1$ relative to other regions. Each action results in rewards following different distribution according to the region to which the given context belongs, where $\mathcal{N}$ denotes the normal distribution.

$$r(1,c) \sim \mathcal{N}(1.2, 0.01^2) \tag{28}$$

$$r(a,c) \sim \begin{cases} \mathcal{N}(50, 0.01^2), & \text{if } c \in R_a \\ \mathcal{N}(1, 0.01^2), & \text{otherwise} \end{cases} \quad \forall a \in \{2,3,4,5\} \tag{29}$$

Note that action $\{1\}$ always produces a moderate reward, but the other actions $\{2,3,4,5\}$ sometimes produce a very high reward when the context is sampled from the corresponding high-reward region. Thus, learning different reward distributions for actions $\{2,3,4,5\}$ by apprehension of context information is critical to bandit performance. As $\delta$ increases, the high-reward regions for each actions become smaller. Then the model should learn from rare observation of such high reward, which means the problem becomes more difficult.

**Training and Evaluation** When pre-training NeuBANP, as in Garnelo et al. (2018b), 8 training batches of 512 contexts and 50 targets were generated from the environment with hyperparameter randomly sampled; $\delta \sim \text{Uniform}(0,1)$. We consider two-dimensional context point $c_i$ as feature $x_i \in \mathbb{R}^2$ and five rewards for actions $(r(1,c_i), r(2,c_i), r(3,c_i), r(4,c_i), r(5,c_i))$ as label $y_i \in \mathbb{R}^5$. At evaluation, only rewards for chosen actions are observed by the model. Thus, we replace other unobserved rewards with *dummy* values randomly sampled from $\mathcal{N}(0,1)$, following the usual strategy.

## B.5 IMAGE COMPLETION

**Settings** We compared the baseline NPs and NeuBANP on image completion tasks. Following Lee et al. (2020), we trained all models on EMNIST (Cohen et al., 2017) and CelebA (Liu et al., 2015) which was resized to $32 \times 32$. For EMNIST, we used only 10 classes for training and reported the evaluation results for both seen classes and unseen classes separately. NeuBANP was trained with 10 samples, and the other baselines were trained with 4 samples. For evaluation, we used 50 samples for all methods. Similar to 1D regression experiment, we randomly select the pixels of a given image as context/target, and the number of context/target were drawn randomly from Uniform distribution. However, in this case, we increased the maximum number of given points; i.e., $|\mathcal{C}| \sim \text{Uniform}(3, 197)$, $|\mathcal{T}| \sim \text{Uniform}(3, 200 - |\mathcal{C}|)$. $x$ values were rescaled to $[-1, 1]$ and the corresponding $y$ values were rescaled to $[-0.5, 0.5]$. We trained all models for 200 epochs and set a initial learning rate of $5 \times 10^{-4}$ using the Adam optimizer (Kingma & Ba, 2015) with cosine annealing scheduler for learning rate decay.

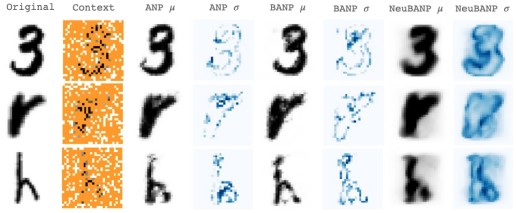

**Fig. 11.** Qualitative result of EMNIST image completion.

**Table 5.** Quantitative result of EMNIST image completion. Mean and standard deviationd of likelihoood over 5 experiments.

| Method | Seen classes (0-9) | | Unseen classes (10-46) | |
|---|---|---|---|---|
| | context | target | context | target |
| CNP | $0.926 _{\pm 0.007}$ | $0.751 _{\pm 0.005}$ | $0.766 _{\pm 0.009}$ | $0.498 _{\pm 0.012}$ |
| NP | $0.948 _{\pm 0.006}$ | $0.806 _{\pm 0.005}$ | $0.808 _{\pm 0.005}$ | $0.600 _{\pm 0.009}$ |
| ANP | $1.383 _{\pm 0.000}$ | $0.993 _{\pm 0.005}$ | $\mathbf{1.383} _{\pm 0.000}$ | $0.894 _{\pm 0.004}$ |
| BNP | $1.004 _{\pm 0.008}$ | $0.880 _{\pm 0.005}$ | $0.883 _{\pm 0.010}$ | $0.722 _{\pm 0.006}$ |
| BANP | $1.383 _{\pm 0.000}$ | $1.010 _{\pm 0.006}$ | $1.382 _{\pm 0.000}$ | $0.942 _{\pm 0.005}$ |
| NeuBANP | $\mathbf{1.475} _{\pm 0.345}$ | $\mathbf{1.337} _{\pm 0.224}$ | $1.333 _{\pm 0.516}$ | $\mathbf{1.119} _{\pm 0.388}$ |

**Fig. 12.** Qualitative result of CelebA image completion.

**Table 6.** Quantitative result of CelebA results. Mean and standard deviationd of likelihoood over 5 experiments.

| | context | target |
|---|---|---|
| CNP | $2.975 _{\pm 0.013}$ | $2.199 _{\pm 0.003}$ |
| NP | $3.066 _{\pm 0.011}$ | $2.492 _{\pm 0.014}$ |
| ANP | $4.150 _{\pm 0.000}$ | $2.731 _{\pm 0.006}$ |
| BNP | $3.269 _{\pm 0.008}$ | $2.788 _{\pm 0.005}$ |
| BANP | $4.149 _{\pm 0.000}$ | $\mathbf{3.129} _{\pm 0.005}$ |
| NeuBANP | $\mathbf{13.946} _{\pm 0.590}$ | $2.870 _{\pm 0.021}$ |

**Results**   Figure 11 and 12 show the mean prediction and uncertainty estimation of ANP, BANP, and NeuBANP for test images in unseen classes. For both datasets, though our model shows noisy mean prediction due to the random weights, we can demonstrate the advantage of NeuBANP in uncertainty estimation. NeuBANP estimated the uncertainty correctly in the area where the color of the pixel changes and thus possesses significant uncertainty. This leads to the overall improvement in quantitative performance (see Table 5 and 6).

## B.6   TIME COMPLEXITY

**Settings**   We measured the time complexity empirically according to the number of context points, target points, and bootstrap samples. We fixed the number of targets to 25 and adjusted the number of contexts to 10, 20, 30, 40, and 50 to see how inference time varies with the number of contexts. Conversely, to see the inference time according to the number of targets, we fixed the number of context points to 25. We fixed the number of bootstrap samples to 50 as in the 1D regression experiment. When conducting experiments with the number of bootstrap samples, the number of context and target points was fixed at 20 and 25. All experiments were conducted with a batch containing 100 tasks.

**Results**   BANP places a remarkably high computational cost in that the approach of bootstrapping the attention module is inefficient, as demonstrated in Figure 13. The inference time becomes noticeably longer as the number of context points increases. NeuBANP, on the other hand, learns to bootstrap efficiently; therefore, its time complexity is comparable to that of BNP, which does not use the attention module.

| Method | Number of contexts | | | | | Number of targets | | | | | Number of bootstrap samples | | |
|---|---|---|---|---|---|---|---|---|---|---|---|---|---|
| | 10 | 20 | 30 | 40 | 50 | 10 | 20 | 30 | 40 | 50 | 10 | 50 | 100 |
| BNP | 1.797 | 1.977 | 2.222 | 2.546 | 2.886 | 1.830 | 1.882 | 1.965 | 2.057 | 2.156 | 1.532 | 1.639 | 1.950 |
| BANP | 3.512 | 4.405 | 5.345 | 6.509 | 7.793 | 4.439 | 4.626 | 4.834 | 4.926 | 5.117 | 3.189 | 3.699 | 4.775 |
| NeuBANP | 1.632 | 1.813 | 2.217 | 2.757 | 3.369 | 1.768 | 1.941 | 2.063 | 2.212 | 2.357 | 1.606 | 1.705 | 2.149 |

**Table 7.**  Inference time measurement. Mean of inference time over 5 runs are reported.

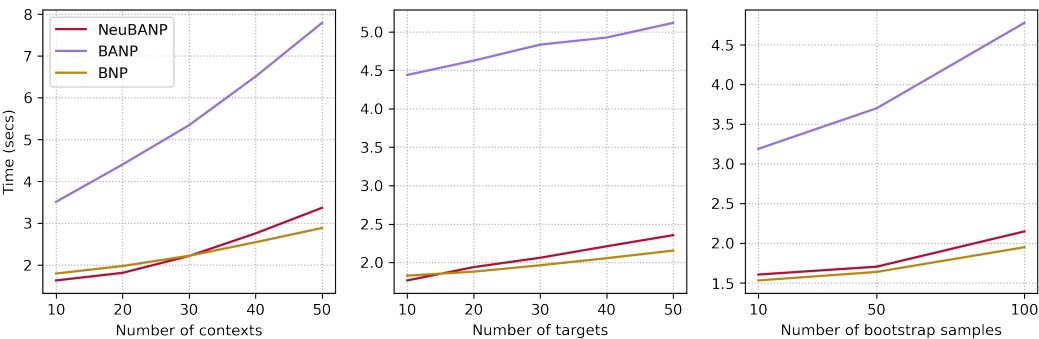

**Fig. 13.** Inference time measurement.

