# OpenReview forum: "Neural Bootstrapping Attention for Neural Processes"
_ICLR.cc/2022/Conference — ICLR 2022 Submitted_

### Official Review · Reviewer_B7VB · 2021-10-28

**Correctness:** 4
**Technical Novelty And Significance:** 2
**Empirical Novelty And Significance:** 3
**Recommendation:** 5
**Confidence:** 3

**Main Review:**

The strength of the paper is that the authors report multiple experiments in different settings that show the superior performance of NeuBANP. They are clear about the comparative advantage from ANP, BANP, and NeuBANP. The figure comparing BANP and NeuBANP makes it easy to understand the difference and highlights the efficiency of NeuBANP. The figures for experimental results also depict the superior performance of NeuBANP well.

My main concern to hold this paper to the high standard of ICLR is the significance and originality of the paper. As it has been discussed well in the paper, Bootstrapping Attentive Neural Processes have been proposed in previous work. Also, Neural Bootstrapper, which replaces the sampling problem of bootstrapping with the augmentation of the loss function, has also been proposed in Shin et al. 2021. This paper looks to me is an application of the ideas in (Shin et al 2021) to the published Bootstrapping Attentive Neural Processes. The paper is well written and the ideas are discussed clearly. However, I am not sure if a combination of existing ideas can win out in a highly competitive venue such as ICLR.

There are some suggestions for the paper as well. First, it would be helpful to mention the significance of NP models compared to other models a little more. Second, one of the contributions the authors mention is computational efficiency. Although this is well depicted in the figure, some more experimental results regarding this point would be better for highlighting this point. Third, if there exist other NP methods apart from ANP and BANP, it might be good to introduce them briefly. Fourth, it would be better if the others made it clearer if the state-of-the-art performance is regarding all other machine learning models or other NP methods. Lastly, including some mathematical details or reasoning of why NeuBANP does better in estimating functional uncertainty would be good.


**Summary Of The Paper:**

The authors propose a new bootstrapping method in NP family called NeuBANP. The authors acknowledge the limitations of BANP and used the neural bootstrapping method to make development from BANP. While BANP does better functional uncertainty estimation than ANP, it carries a higher computational burden compared to ANP since it requires multiple computations of the encoder network, the adaptation layer, and additional heuristics. By incorporating neural bootstrapping to the bootstrapping procedure, NeuBANP is a more computationally efficient model, estimates functional uncertainty better than BANP, and also alleviates the overfitting problem that ANP and BANP carries. Experimental results show that it achieves state-of-the-art performance on stochastic optimization problems, including multidimensional Bayesian optimization and contextual multi-armed bandit.

**Summary Of The Review:**

The authors propose a new bootstrapping method in NP family called NeuBANP. The authors acknowledge the limitations of BANP and used the neural bootstrapping method to make development from BANP. The experimental results show great improvement over other NP methods. While the paper is written well and the figures are clear, there is some room for improvement. In particular, the combination of existing ideas may not hold up to the high standard of ICLR.

---

> ### Author Response · Authors · 2021-11-16
> **Replies to Reviewer B7VB by Authors**
>
> *Thank you for the helpful comments. Following is the list of our answers to your concerns and the corresponding updates on our revised manuscript.*
>
> **1. Concern on the originality**
>
> It is true that B(A)NP opens the possibility of using bootstrap into NP models. However, our method is completely separate one from B(A)NP because our bootstrap method is applied to ANP. It would be helpful to understand if you to think of our method and B(A)NP as parallel strategies to introduce bootstrapping in NP models. Also, our method is different from simple combination of ANP and NeuBoots, in three points.
>
> First, We concatenate random bootstrap weights to context data, unlike NeuBoots, which only utilizes random weight multiplication in the final layer. Effect of this modification is explained in the paragraph about Prediction path in Section 3.1.
>
> Second, we only apply weighted loss on context data points, whereas NeuBoots apply it on every training data points. You can find this point in Section 3.2.
>
> Third, the output of NeuBANP is different from naive application of NeuBoots on ANP. ANP outputs $\mathcal{N}(\mu, \sigma^2)$. NeuBoots estimate nonparametric bootstrap distribution with many outputs $\hat y^{(b)}$. So if one uses both strategy at once, the output would be $\frac{1}{B}\sum_{b=1}^B\mathcal{N}(\mu^{(b)},{\sigma^{(b)}}^2)$. However, we found that this does not work empirically. We decided to generate the sample outputs $\hat y^{(b)}$ instead of the density network output as in ANP, and we estimate the predictive distribution $\mathcal{N}(\hat \mu, \hat \sigma^2)$ with the sample statistics $\hat \mu, \hat \sigma$ of sample outputs. We do not use the density network output as in other NPs. We think this point is well elaborated in the paragraph about Decoder and Bootstrapped Prediction in Section 3.1.
>
> *The authors sincerely appreciate the comments from reviewers. The followings are our responses to the reviewer’s suggestions.*
>
> **1. Significance of NP models compared to other models**
>
> First, NP models can utilize various neural network architectures for its encoder and decoder network.
>
> Second, NP models enjoy linear computational complexity. Using attention comes at a cost of square complexity, but still faster than GP.
>
> Third, NP models do not have to choose which kernel to use and corresponding kernel parameters, because it is designed to learn appropriate kernel function implicitly in data-driven way. This advantage becomes stronger with the size of training dataset.
> We think these points are well elaborated on the previous NP works, but thanks to your suggestion, we added more examples of NPs in Section 5 of our paper (Related Work) to show the significance of NP models.
>
> **2. Experimental results on the computational efficiency**
>
> Thanks for your suggestion, we added the comparison of inference time of BNP, BANP, and NeuBANP in the new section (Appendix B.6). In summary, NeuBANP and BNP have similar inference time, while BANP has more than twice longer inference time.
>
> **3. Introduction of other NP methods**
>
> To inform more about NP methods, we added some of them in Section 5 (Related Work).
>
> **4. SOTA regarding all machine learning models, or NP methods?**
>
> Our method is SOTA regarding NP methods. Thanks for your suggestion, we made it clear in the revised abstract.
>
> **5. Mathematical details or reasoning of why NeuBANP does better?**
>
> We argue this in three points.
>
> First, since NeuBoots used in NeuBANP is theoretically proven to be asymptotically approximated as standard bootstrap, it is also possible to measure uncertainty nonparametrically through bootstrapping in NeuBANP.
>
> Second, standard bootstrap has a data discarded problem, but because NeuBoots samples random weights from Dirichlet distribution, the problem fundamentally disappears because NP learns how to bootstrap by adjusting the gradient weight of each data point. The bootstrap method used in BNP/BANP used both residual bootstrap and paired bootstrap to bypass the data discarded problem, but they introduced an additional adaptation layer for robust performance.
>
> Third, in the case of BNP/BANP, even though bootstrapping is used, variance estimation is used as the output of the decoder as shown in Equation (4), and uncertainty is learned by adding a lower bound, so accurate estimation is not possible. However, in the case of NeuBANP, variance is estimated with bootstrap samples and predictive mean as in Equation (10), and since a lower bound is not used, it is unbiased and estimates variance accurately.

---

### Official Review · Reviewer_1P97 · 2021-11-01

**Correctness:** 3
**Technical Novelty And Significance:** 3
**Empirical Novelty And Significance:** 3
**Recommendation:** 6
**Confidence:** 3

**Main Review:**

Strength:

- Problem of modeling functional uncertainty is clearly motivated.
- Simplifies and improves BANP using NeuBoots.
- Evaluates on standard benchmarks and shows superior results on Bayesian optimization and contextual multi-armed bandit tasks.
- Provides clean structured code for reproducibility.

Weakness:

- The empirical evaluation on 1D regression remains mixed. It’s hard to tell whether the failure of modeling GP target samples from the periodic kernel (Table 2) is intrinsic to the proposed method or if the experiment is not executed properly without having further analysis. Also it seems like context LL is significantly better for regression tasks but that has not been explained enough.

- One of slight worry from the reviewer is that the paper’s method is hinging on two of very recently proposed approaches (Bootstrapping Attentive Neural Process and Neural Bootstrapper) where each of the merits are not quite validated thoroughly in the literature yet.


Suggestions:

- Main interesting idea this paper brings into NP space is utilizing Neural Bootstrapper. However, current analysis heavily relies on building on BANP. It would be great ablation analysis to apply neural bootstrapper on non-attentive models and compare to vanilla BNP and NPs.  While the idea of using attentive neural processes may fade-away,  the idea of efficient bootstrapping of neural processes could remain important.

- Figure 1 / Figure 6 Training setting is unclear and makes it hard to interpret what message is being delivered.

- In section 4.1. 1D regression seems cherry-picked since it only shows the RBF kernel and does not discuss failure in the Periodic (in Appendix) while standard 1D evaluation runs on various kernels (RBF, Matern, Periodic).

- Section 4.3 results on CMAB are quite promising and demonstrate competitiveness. It would be great to include a comparison to BANP and ANP as well to see that Neural Bootstrapping is the main reason for superior performance.

- In Figure 5, it would be interesting to have a sense of order of evaluated points to see how each model zooms into particular (often suboptimal) minima.

Questions:
- As mentioned earlier, do the authors have a good idea why improvements on the context point are huge? According to the motivation NeuBANP was introduced to solve ANPs “overfitting” issue. Having high context LLs doesn't quite align with the motivation?

- In Figure 3, NeuBANP samples as well as posterior distribution are discontinuous; why is this the case? (e.g around -0.5, and 0.5) With the discontinuity, it does not seem like a great model but still able to obtain good context/target LLs?

- In Table 5, ANP’s context LL for both seen classes and unseen classes is exactly the same, did authors check that if there’s mistake on reporting or the actual numbers turned out to have the same mean and std.

- While authors cite Lakshminarayanan et al. 2017 for bootstrapping to be a reliable approach to estimate predictive uncertainty in neural networks, the reviewers take from that paper is that bootstrapping was detrimental compared to deep ensemble. In Nixon et al., 2020 (https://openreview.net/forum?id=dTCir0ceyv0), they further observe bootstrapping doesn't improve uncertainty beyond the vanilla ensemble method. Could authors elucidate in what situation we expect using bootstrapping to be beneficial vs not in neural networks?

- Caption in Table2: is reporting on the 48,000 evaluation standard procedure? seems quite arbitrary and high


------------------------------------------------------------------------------

post rebuttal : I thank the author for the response and clarification. My assessment still is that this is somewhat borderline paper for ICLR and slightly leaning supporting acceptance thanks to solid result on sequential decision making tasks.

**Summary Of The Paper:**

The paper proposed a new class of neural process algorithm called neural bootstrapping attention for neural processes (NeuBANP). This method utilizes efficient Neural Bootstrapping (NeuBoots) to improve Bootstrapped Attentive Neural Processes (BANP) in capturing functional uncertainty. Authors show that NeuBANP achieves state-of-the-art performance in benchmark experiments including Bayesian optimization and contextual multi-armed bandits.


**Summary Of The Review:**

This paper proposes an efficient way of modeling functional uncertainty building on recent work of NeuBoots and BANP. On the benchmark of sequential decision making, authors demonstrated state-of-the art performance which shows great promise. Method is presented in a clear fashion and evaluation is done on standard set up. There are some desired analyzes pointed out in the main review that could improve scientific understanding of the proposed method. Overall, I believe the proposed idea is sound and interesting to be shared among the ICLR audience.

---

> ### Author Response · Authors · 2021-11-16
> **Replies to Reviewer 1P97 by Authors (2)**
>
> *The authors sincerely appreciate the comments from reviewers. The followings are our responses to the reviewer’s suggestions.*
>
> **1. Ablation study on non-attentive model**
>
> Thanks for your thoughtful suggestion. We designed the non-attentive counterpart of our method, named Neural Bootstrapping Neural Processes (NeuBNP). We added the explanation of its architecture in Appendix A.3, and added the experimental results to Table 2 (1d regression) and Table 3 (1d BO). In summay, NeuBNP performs worse than NeuBANP, but outperforms BNP and NP, showing the quality of uncertainty estimation learned by our bootstrap method.
>
> **2. Training setting and the message in Figure 1 and 6**
>
> In Figure 1 and 6, we experimented using a linear function rather than a function sampled from the GP to analyze how well the NP models work in a simple regression task. Additionally, we need to examine how well the NP models predict uncertainty in the presence of heterogeneous noise in the data, and we add $\epsilon(x) \sim \mathcal N(0, \sigma^2(x)), \sigma(x)= \sqrt{x^2 + 10^{-5}}$ to the linear function. We multiplied the noise by coefficient $\beta$ to follow the meta-learning framework, and the $\beta$ value was uniformly randomly sampled from $[0.1, 1.0]$ during training. In this task, NP, BNP, and NeuBANP were able to estimate the underlying true linear function and the uncertainty of heterogeneous noise. On the other hand, in case of ANP and BANP, overfitting occurred and failed to predict the true function and its uncertainty (see Figures 1 and 6). As explained in the main text, we can see that the overfitting phenomenon occurs because of the attention mechanism that has appeared to solve the underfitting issue of the NP. NeuBANP has the advantage of the attention mechanism and achieved better performance through regularization using random weights. The remaining training settings are the same as the 1D regression in Appendix B.2, and we changed only the training iteration to 10,000. We added this explanation in Appendix B.1.
>
> **3. Comparison of BANP and ANP on CMAB experiment**
>
> We added the result of BANP and ANP on CMAB experiment in Table 1, and added an explanation of this result in Section 4.3. In summary, NeuBANP outperforms BANP and ANP, especially when the bandit environment becomes more difficult.
>
> **4. Posterior distribution discontinuity issue**
>
> Fisrt of all, this phenomena commonly occurs for attentive models (ANP, BANP, NeuBANP). The non-smoothness of mean prediction may seem defective, but we want to argue that this occurs in a small window of the domain space, and this can be recovered by largely estimated uncertainty. Thus, despite this limitation of attentive NPs, our model shows the best performance in average with the help of better uncertainty estimation than others, as you can see in the log-likelihood results. We added an explanation for this in the last paragraph of Appendix B.2.
>
> **5. Numerical Issues**
>
> > ANP's context LL is exactly same in Table 5
>
> We understand that it may seem doubtful, but this is the same as the official result reported by BNP paper (Lee et al., 2020, Table 2).
>
> > Caption in Table 2, reporting on the 48,000 evaluations
>
> We followed the experimental setting of BNP as a standard, or benchmark, and the number of evaluations is also borrowed from their work.
>
> **6. When bootstrapping is beneficial in neural networks?**
>
> In general, bootstrapping is beneficial in few-shot setting, like the early stages in sequential decision making, where only a small number of evaluated points are given. When you bootstrap the training dataset with paired bootstrapping, some of the data will be discarded, and the model trained with such bootstrapped training dataset will perform worse than the model trained with the full training dataset. We call this the data missing data problem of bootstrapping. However, NeuBoots is beneficial in any situation, because this method has no need to resample the dataset, so does not suffer from the missing data problem.

---

> ### Author Response · Authors · 2021-11-16
> **Replies to Reviewer 1P97 by Authors (1)**
>
> *Thank you for the helpful comment. Following is the list of our answers to your concerns and the corresponding updates on our revised manuscript.*
>
> **1. Empirical evalutation on 1D regression with different kernels**
>
> We added figures showing the predictions of ANP, BANP, and NeuBANP for the Matern kernel and Periodic kernel in Appendix B.2. In the case of the Matern kernel, the two problems described in the main text occurred identically for ANP and BANP (see Section 4.1 Results). However, in the case of Periodic, we can see that all models failed to approximate the true function correctly. This result came out because we experimented with testing the models' generalization performance when trained with the RBF kernel. And the quantitative results in Table 2 show that the prediction performance of the attention-based model on the target data is poor. Among them, NeuBANP has the worst performance, and we think the reason is the lower bound on the predicted variance set by ANP and BANP. Quantitative results show similar predictions for all models, but ANP and BANP achieve numerically more robust performance by setting a lower bound on the variance.
>
> **2. Explanation of high context LL in 1d regression**
>
> The key reason for the higher performance of NeuBANP on context points, we believe, is that the output is defined differently from previous NP models, and the predicted variance has no lower bound. Except in the case of the Periodic kernel, NeuBANP has the best performance even for target points. The generalization performance of NP is to accurately predict a function not given in training since it operates in a meta-learning framework. Therefore, when looking at the results of the regression task in the Matern kernel, Bayesian optimization for several benchmark functions, and contextual multi-armed bandit experiments, we believe that the generalization performance of NeuBANP is better. And in the simple regression task (see Figure 1), unlike NeuBANP, overfitting obviously occurred in ANP and BANP, so the random weight utilized in NeuBANP brings a regularization effect, empirically demonstrating that it has better generalization performance.
>
> **3. Our method hinging on B(A)NP and NeuBoots**
>
> It is true that B(A)NP opens the possibility of using bootstrap into NP models. Although B(A)NP proposed applying the bootstrap method to NP for the first time, we found that it did not solve the problems of overfitting and uncertainty estimation that occur in ANP. In addition, BANP introduced a computationally heavy architecture to obtain robust performance using the bootstrap method, and this model brings the additional problem that it does not model functional uncertainty well. To solve this problem, we introduced a new bootstrap method, NeuBoots, to ANP. By proposing a new NP family, NeuBANP, we achieved superior performance in the regression and black-box optimization tasks compared to the existing NP methods. Also, even though we borrowed the idea from NeuBoots, our problem setting is different from NeuBoots. NeuBoots is trained in simple learning setting, mainly on image classification and segmentation, however, our method is trained in meta-learning setting to learn random functions from stochastic processes. We think NeuBoots has given concrete justifications of their method in their paper, but regardless of that, we think our experimental results are persuasive enough to show the effect of our bootstrapping method in NP setting.

---

### Official Review · Reviewer_Me1R · 2021-11-02

**Correctness:** 3
**Technical Novelty And Significance:** 2
**Empirical Novelty And Significance:** 3
**Recommendation:** 5
**Confidence:** 3

**Main Review:**

Strength:
- The paper is written in clear manner.
- The proposed method is easy-to-implement in NP schemes.
- The code is also provided together for the reproducibility.

Weakness:
- The novelty is somewhat limited, that this is a direct utilization of Neural Bootstrapper.
- I'm confused on the message of the paper. In Fig 1, the authors argue that the baselines (1) overfits the data from the linear regression samples, and (2) are incapable of capturing uncertainty. However, in Fig 3, the proposed NeuBANP seems overfitted and the baselines seems that they can capture the uncertainty (at least) in interpolation. If I missed something, please let me know.




**Summary Of The Paper:**

The paper proposes Bootstrapping Attentive Neural Processes (NeuBANP), which utilize random sum-to-one weight in the encoder following from Neural Bootstrapper (Shin et al., 2021). The authors argue that this utilization resolves the overfitting problem in attentive NPs, and the proposed NeuBANP is more efficient in computation and memory perspective. Various experiments, such as synthetic examples and contextual multi-armed bandit, are conducted to compare against previous works of NPs and GP.

**Summary Of The Review:**

My opinion is that the paper is on borderline. The work seems neat and probable while the novelty is limited. Considering the quality of the venue, I'm bit negative on the direct utilization of existing work. Also, there is a disalignment (probably my misunderstanding) of the motivation and the experiment result, which I discussed in the weakness, want to be addressed.

---

> ### Author Response · Authors · 2021-11-16
> **Replies to Reviewer Me1R  by Authors**
>
> *Thank you for the helpful comment. Following is the list of our answers to your concerns and the corresponding updates on our revised manuscript.*
>
> **1. Concern on the novelty.**
>
> We added some explanation about the novelty of our method in the revised manuscript, but we want to explain the main point here. It is true that our method enjoys the simplicity and computational efficiency of NeuBoots. However, this is not a straightforward combination of ANP and NeuBoots in three points. First, We concatenate random bootstrap weights to context data, unlike NeuBoots, which only utilizes random weight multiplication in the final layer. Effect of this modification is explained in the paragraph about Prediction path in Section 3.1. Second, we only apply weighted loss on context data points, whereas NeuBoots apply it on every training data points. You can find this point in Section 3.2. Third, the output of NeuBANP is different from naive application of NeuBoots on ANP. ANP outputs $\mathcal{N}(\mu, \sigma^2)$. NeuBoots estimate nonparametric bootstrap distribution with many outputs $\hat y^{(b)}$. So if one uses both strategy at once, the output would be $\frac{1}{B}\sum_{b=1}^B\mathcal{N}(\mu^{(b)},{\sigma^{(b)}}^2)$. However, we found that this does not work empirically. We decided to generate the sample outputs $\hat y^{(b)}$ instead of the density network output as in ANP, and we estimate the predictive distribution $\mathcal{N}(\hat \mu, \hat \sigma^2)$ with the sample statistics $\hat \mu, \hat \sigma$ of sample outputs. We do not use the density network output as in other NPs. We think this point is well elaborated in the paragraph about Decoder and Bootstrapped Prediction in Section 3.1.
>
> **2. Confusion on the message of the paper, regarding Figure 1 and 3.**
>
> In Figure 1, we experimented using a linear function rather than a function sampled from the GP to analyze how well the NP models work in a simple regression task. Additionally, we need to examine how well the NP models predict uncertainty in the presence of heterogeneous noise in the data, and we add $\epsilon(x) \sim \mathcal N(0, \sigma^2(x)), \sigma(x)= \sqrt{x^2 + 10^{-5}}$ to the linear function. We multiplied the noise by coefficient $\beta$ to follow the meta-learning framework, and the $\beta$ value was uniformly randomly sampled from $[0.1, 1.0]$ during training. In this task, NP, BNP, and NeuBANP were able to estimate the underlying true linear function and the uncertainty of heterogeneous noise. On the other hand, in case of ANP and BANP, overfitting occurred and failed to predict the true function and its uncertainty (see Figures 1 and 6). As explained in the main text, we can see that the overfitting phenomenon occurs because of the attention mechanism that has appeared to solve the underfitting issue of the NP. NeuBANP has the advantage of the attention mechanism and achieved better performance through regularization using random weights. The remaining training settings are the same as the 1D regression in Appendix B.2, and we changed only the training iteration to 10,000.
> However, in Figure 3, we described two problems (see Section 4.1 Results). One problem occurred in ANP and BANP, estimating the identical uncertainty for all intervals in a situation where context points are sufficiently given. Another problem only occurs in BANP, which is that it does not properly estimate functional uncertainty. Since the uncertainty about the shape of the true underlying function is high in the region where the context point is not given, models should generate a wide range of function samples. However, we can see that BANP generates almost the same functions, unlike ANP and NeuBANP, and therefore, we explained that the residual bootstrap is not an appropriate method for modeling functional uncertainty. Since we trained the models in Figure 3 from a much more complex function (random function from GP) than Figure 1, overfitting did not occur in all models, and we added Figure 3 to explain the two problems pointed out above qualitatively.
> We added this explanation in Appendix B.1. Additionally, we revised the explanation about Figure 3 to be more clear, in Section 4.1.

---

### Official Review · Reviewer_fWXh · 2021-11-02

**Correctness:** 3
**Technical Novelty And Significance:** 2
**Empirical Novelty And Significance:** 2
**Recommendation:** 6
**Confidence:** 4

**Main Review:**

Positive aspects:
* Overall the idea of extending the bootstrapping element of B(A)NPs using neural bootstrapping makes sense and seems to work well in practice.
* The authors do provide a nice set of experiments with the appropriate baselines and the results seem promising.


Criticisms:
* My main concern is that it is not clear to me why one would expect that applying this neural version of bootstrapping would improve the uncertainty prediction of the model and allow for the heteroscedastic uncertainty prediction. From the results it seems that this is indeed the case, but it would be useful to have some intuition behind this explained in the paper. The way I understand the paper this is the main benefit of this model upgrade so this should be elaborated on.
* The second benefit of this model seems to be that it is computationally more efficient (given that it does not require the iterative bootstrapping step, this makes sense). It might be worth analysing how much this buys you in terms of computation/time. Given that this is one of the two selling points it would be nice to have a more quantitative evaluation of this.
* Finally, in terms of novelty, while this is the first time I hear of this particular combination of models, it is a somewhat straightforward combination of previously existing ideas (this in itself should not render the work unworthy of being published, of course).

Suggestions:
* I personally find the notation used to state that you are sampling the context points  $C$ from some set of indices $[n]$ a bit confusing. In equation 1 the inner expectation is wrt C but then the conditional probability is on $D_C$. I would personally write it as: $E_{C \sim D_C}-\log p_{\theta}(Y|X, C)$.
* In the paragraph on BANP I would define the variable $b$ much earlier (right when it is first used in the fourth line) rather than waiting to the end, as it is not clear that these are the different ensemble samples.
* In figure 2 I would make clear that both the bootstrapping path in B(A)NP and the encoding part in NeuBANP is repeated b times (if I understood correctly this is only shown for one instance in the figure).
* Also in figure 2 I would add the $\hat{r}$ variable in the diagram just before the decoder, since you mention it in the text.
* In figure 3 I would explicitly mention that the minimum variance of ANP and BANP is as large as it appears because that is the fixed minimum value. Not to say that there is no benefit from NeuBANP not having such a lower bound, but for those that might not be familiar with that fixed value in the other models the results might seem odd.


**Summary Of The Paper:**

The authors introduce NeuBANP, an extension of B(A)NP that replaces the iterative prediction method of B(A)NP with a single prediction step following the idea behind the neural bootstrapper. This new model is computationally more efficient and able to produce more accurate uncertainty estimates as well as correctly model heteroscedastic uncertainties. The authors provide experimental results on nonparametric regression tasks, Bayesian optimisation, contextual multi-armed bandit tasks and image inpainting (in the appendix).

**Summary Of The Review:**

Overall I think the paper is nice, the explanations are not hard to follow and it is overall well written. While the novelty of this model is not outstanding, the results section covers an interesting range of experiments with the right baselines. I would personally recommend adding more intuition as to why this extension improves the performance the way it does, as it is the main selling point.

---

> ### Author Response · Authors · 2021-11-16
> **Replies to Reviewer fWXh by Authors (2)**
>
> *The authors sincerely appreciate the comments from reviewers. The followings are our responses to the reviewer’s suggestions.*
>
> **1. Notation of sampling distribution for the context index set $\mathcal{C}$**
>
> In our knowledge, people use $\mathcal{C} \sim [n]$ as an abbreviation of sampling random subsets from $\{ 1,2,...,n \}$. However, we understand that not everyone is familiar to this kind of notation. So we changed this to another expression $\mathcal{C} \sim P_n$, and gave explanation about the sampling distribution $P_n$. You can see this change in Section 2.1.
>
> **2. Explanation of $b$, the bootstrap index.**
>
> In the revised manuscript, we explain this right after this notation appears, for better readability.
>
> **3. Mark B times of repetitions in Figure 2**
>
> As we mentioned on the caption, this figure shows the single forward computation paths. Thanks for your suggestion, but we decided to not change the figure. Instead, we mentioned in the caption that this procedure must be repeated for B times.
>
> **4. Mark $\hat r$ before the decoder in Figure 2**
>
> Forward calculation of BANP has two steps. First step is to construct the residual bootstrap samples, and the second is to calculate the prediction. Each steps corresponds to the left and right part of BANP in Figure 2. Since $\hat r$ only appears in the second step, as explained in Section 2.2, $\hat r$ should only appear on the right part of BANP in Figure 2. So there will be no change in Figure 2.
>
> **5. Explicitly mention the fixed minimum value on the variance in Figure 3**
>
> We added this point in the paragraph explaining the results of 1d regression, in Section 4.1.

---

> ### Author Response · Authors · 2021-11-16
> **Replies to Reviewer fWXh by Authors (1)**
>
> *Thank you for the helpful comment. Following is the list of our answers to your concerns and the corresponding updates on our revised manuscript.*
>
> **1. Intuition behind why our method works well in uncertainty prediction**
>
> To represent the functional uncertainty behind the data generating process like GP, we need a global randomness in the model that results in somewhat random but consistent predictions so they come from one function at a time. In ANP the latent path stands for this. BANP shows the possibility of using bootstrapping to induce such randomness. Now, our proposition is that learning to bootstrap is better than the fixed bootstrap as in BANP. In BANP, bootstrapping procedure (paired bootstrap + residual bootstrap) does not change depending on the context. On the other hand, our bootstrap strategy learns how to construct bootstrap distribution depending on the context. We tag the bootstrap weight to each context, and the model process this weights into corresponding bootstrap sample predictions. Optimizing the weighted loss function (Section 3.2) updates the model parameter considering the functional uncertainty estimated by such bootstrapped predictions calculated by the model. This learning strategy guides the model to select how to construct the functional uncertainty using the given context and learned general property of the underlying stochastic processes that made the training dataset. We added the main point of this explanation in the first paragraph of Section 3. Finally, we think this is a proper step to improve bootstrap methods for NPs as the authors of BNP paper (Lee et al., 2020) also wrote about it in the conclusion (“Designing a framework that could “learn” to resample bootstrap datasets in a data-driven way would be an interesting and promising research direction”).
>
> **2. Quantitative evaluation on computational efficiency**
>
> We agreed that we should give experimental results about efficiency of models. So we added the comparison of inference time of BNP, BANP, and NeuBANP in the new section (Appendix B.6). In summary, NeuBANP and BNP have similar inference time, while BANP has more than twice longer inference time.
>
> **3. Why NeuBANP is not straightforward combination of ANP and NeuBoots**
>
> Our method is different from simple combination of ANP and NeuBoots, in three points.
>
> First, We concatenate random bootstrap weights to context data, unlike NeuBoots, which only utilizes random weight multiplication in the final layer. Concatenation of weights and the contexts in the posterior path yields two effects. First, concatenating random information in given contexts provides sufficient randomness to the model, allowing it to cover a wide range of function samples in a space where a true function is likely to exist. Second, we made this modification to maximize the use of random weights and provide bootstrapping information to the model without resampling the data, enabling better uncertainty estimation. This modification is added in the paragraph about Prediction path in Section 3.1.
>
> Second, we only apply weighted loss on context data points, whereas NeuBoots apply it on every training data points. You can find this point in Section 3.2.
>
> Third, the output of NeuBANP is different from naive application of NeuBoots on ANP. ANP outputs $\mathcal{N}(\mu, \sigma^2)$. NeuBoots estimates nonparametric bootstrap distribution with many outputs $\hat y^{(b)}$. So if one uses both strategy at once, the output would be $\frac{1}{B}{\sum}_{b=1}^B\mathcal{N}({\mu}^{(b)},{{\sigma}^{(b)}}^2)$. However, we found that this does not work empirically. We decided to generate the sample outputs $\hat y^{(b)}$ instead of the density network output as in ANP, and we estimate the predictive distribution $\mathcal{N}(\hat \mu, \hat \sigma^2)$ with the sample statistics $\hat \mu, \hat \sigma$ of sample outputs. We do not use the density network output as in other NPs. We think this point is well elaborated in the paragraph about Decoder and Bootstrapped Prediction in Section 3.1.

---

### Decision · Program_Chairs · 2022-01-20

**Decision:**

Reject

**Comment:**

Overall, the work is borderline with no reviewer feeling strongly for or against the paper.

The paper is well-written and proposes a simple approach, along with code for reproducibility. Criticism stems primarily in the work's technical novelty, being an incremental improvement of ideas from ANP and BANP, and related work like Neural Bootstrapper. In addition, the experimental validation involves regression on 1-to-2D functions, Bayesopt on synthetic functions, and contextual bandits on the synthetic wheel bandit problem. This is fairly toy, and multiple reviewers raise unaddressed concerns on the regression experiments. Ignoring orginality in and of itself (which is overvalued in conferences), the work does not yet provide a sufficiently convincing demonstration of its practical importance.

I recommend the authors use the reviewers' feedback to enhance their preprint should they aim to submit to a later venue.